# Fast and Furious Learning in Zero-Sum Games: Vanishing Regret with Non-Vanishing Step Sizes

**James P. Bailey**
Texas A&M University
jamespbailey@tamu.edu

**Georgios Piliouras**
Singapore University of Technology and Design
georgios@sutd.edu.sg

## Abstract

We show for the first time that it is possible to reconcile in online learning in zero-sum games two seemingly contradictory objectives: vanishing time-average regret and non-vanishing step sizes. This phenomenon, that we coin "fast and furious" learning in games, sets a new benchmark about what is possible both in max-min optimization as well as in multi-agent systems. Our analysis does not depend on introducing a carefully tailored dynamic. Instead we focus on the most well studied online dynamic, gradient descent. Similarly, we focus on the simplest textbook class of games, two-agent two-strategy zero-sum games, such as Matching Pennies. Even for this simplest of benchmarks the best known bound for total regret, prior to our work, was the trivial one of $O(T)$, which is immediately applicable even to a non-learning agent. Based on a tight understanding of the geometry of the non-equilibrating trajectories in the dual space we prove a regret bound of $\Theta(\sqrt{T})$ matching the well known optimal bound for adaptive step sizes in the online setting. This guarantee holds for all fixed step-sizes without having to know the time horizon in advance and adapt the fixed step-size accordingly.As a corollary, we establish that even with fixed learning rates the time-average of mixed strategies, utilities converge to their exact Nash equilibrium values. We also provide experimental evidence suggesting the stronger regret bound holds for all zero-sum games.

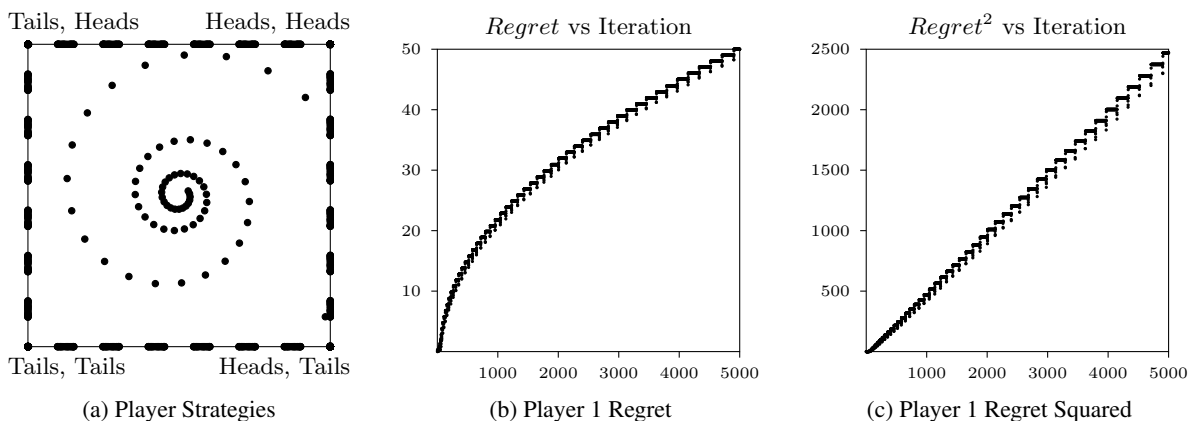

(a) Player Strategies      (b) Player 1 Regret      (c) Player 1 Regret Squared

Figure 1: 5000 Iterations of Gradient Descent on Matching Pennies with $\eta = .15$.

# 1 Introduction

The performance of online learning algorithms such as online gradient descent in adversarial, adaptive settings is a classic staple of optimization and game theory, e.g, Cesa-Bianchi and Lugoisi [2006], Fudenberg and Levine [1998], Young [2004]. Arguably, the most well known results in this space are the following:

- i) Sublinear regret of $O(\sqrt{T})$ is achievable in adversarial settings but only after employing a carefully chosen sequence of shrinking step-sizes or if the time horizon is finite and known in advance and the fixed learning rate is selected accordingly.
- ii) Sublinear regret algorithms "converge" to Nash equilibria in zero-sum games.

Despite the well established nature of these results recent work has revealed some surprising insights that come to challenge the traditional ways of thinking in this area. Specifically, in the case of zero-sum games what is referred to as "convergence" to equilibrium, is the fact that when both agent apply regret-minimizing algorithms, both the time-average of the mixed strategy profiles as well as the utilities of the agents converge approximately to their Nash equilibrium values, where the approximation error can become arbitrarily close to zero by choosing a sufficiently small step-size. Naturally, this statement does not imply that the day-to-day behavior converges to equilibria. In fact, the actual realized behavior is antithetical to convergence to equilibrium. Bailey and Piliouras [2018] showed that *Nash equilibria are repelling in zero-sum games* for all follow-the-regularized-leader dynamics. As seen in Figure 1 the dynamics spiral outwards away from the equilibrium.

These novel insights about the geometry of learning dynamics in zero-sum games suggest a much richer and not well understood landscape of coupled strategic behaviors. They also raise the tantalizing possibility that we may be able to leverage this knowledge to prove tighter regret bounds in games. In fact, a series of recent papers has focused on beating the "black-box" regret bounds using a combination of tailored dynamics and adaptive step-sizes, e.g, Daskalakis et al. [2011], Rakhlin and Sridharan [2013], Syrgkanis et al. [2015], Foster et al. [2016] but so far no new bounds have been proven for the classic setting of fixed learning rates. Interestingly, Foster et al. [2016] explicitly examine the case of fixed learning rates $\eta$ to show that learning achieves sublinear "approximate regret" where the algorithm compares itself against $(1 - \eta)$ times the performance of the best action with hindsight. In contrast, our aim is to show sublinear regret for fixed $\eta$ using the standard notion of regret.

Intuitively, non-equilibration and more generally this emergent behavioral complexity seem like harbingers of bad news in terms of system performance as well as of significant analytical obstacles. This pessimism seems especially justified given recent results about the behavior of online dynamics with fixed step-sizes in other small games (e.g. two-by-two coordination/congestion games), where their behavior can be shown to become provably chaotic (Palaiopanos et al. [2017], Chotibut et al. [2018]). Nevertheless, we show that we can leverage this geometric information to provide the first to our knowledge sublinear regret guarantees for online gradient descent with fixed step-size in games. Instability of Nash equilibria is not an obstacle, but in fact may be leveraged as a tool, for proving low regret.

**Our theoretical results.** We study the dynamics of gradient descent with fixed step size in two-strategy, two-player games. We leverage a deep understanding of the geometry of its orbits to prove the first sublinear regret bounds despite the constant learning rate. We show that the player strategies are repelled away from the Nash equilibrium. More specifically, regardless of the choice of the initial condition there are only a finite number of iterations where both players select mixed strategies (Theorem 1). We prove a worst-case regret bound of $O(\sqrt{T})$ for arbitrarily learning without prior knowledge of $T$ (Theorem 3) matching the well known optimal bound for adaptive learning rates. An immediate corollary of our results is that time-average of the mixed strategy profiles as well as the utilities of the agents converge to their *exact* Nash equilibrium values (and not to approximations thereof) (Corollary 4). Finally, we present a matching lower bound of $\Omega(\sqrt{T})$ (Theorem 5) establishing that our regret analysis is tight.

To obtain the upper bound, we establish a tight understanding of the geometry of the trajectories in the dual space, i.e., the trajectories of the payoff vectors. We show there exists a linear transformation of

the payoff vectors that rotate around the Nash equilibrium. Moreover, the distance between the Nash equilibrium and these transformed utility vectors increases by a constant in each rotation (Lemma 8). In addition, the time to complete a rotation is proportional to the distance between the Nash equilibrium and the transformed payoff vectors (Lemma 9). Together, these results imply a quadratic relationship between the number of iterations and the number of rotations completed establishing the $O(\sqrt{T})$ regret bound. We establish the lower bound by exactly tracking the strategies and regret for a single game.

**Our experimental results.** Many of the proof techniques we develop extend to higher dimensions suggesting sublinear regret in general zero-sum games. To test this, we conducted experiments to measure regret in higher dimension. Our simulations for 5x5, 10x10, and 50x50 games suggest that regret is sublinear and close to $\Theta(\sqrt{T})$ for larger games. A summary of our simulations are given in Table 1 and the fully details appear in Appendix I.

Table 1: Regression Summary for 10,000 Iterations of Gradient Descent in 30 Random Games

| strategies | $Regret_1(T) \approx b \cdot T^a$ | p-value | % of variability explained | \|support of $x^*$\| |
|---|---|---|---|---|
| 2 | $a \in [0.4492, 0.5248]$ | $< .000001$ | $93.53403 - 99.83818$ | 2 |
| 5 | $a \in [0.3662, 0.5504]$ | $< .000001$ | $97.04427 - 99.91377$ | 2-5 |
| 10 | $a \in [0.4653, 0.5563]$ | $< .000001$ | $98.79963 - 99.87485$ | 3-7 |
| 50 | $a \in [0.5260, 0.5776]$ | $< .000001$ | $99.40158 - 99.86970$ | 21-30 |

## 2 Preliminaries

A two-player game consists of two players $\{1, 2\}$ where each player has $n_i$ strategies to select from. Player $i$ can either select a pure strategy $j \in [n_i]$ or a mixed strategy $x_i \in \mathcal{X}_i = \{x_i \in \mathbb{R}_{\geq 0}^{n_i} : \sum_{j \in [n_i]} x_{ij} = 1\}$. A strategy is fully mixed if $x_i \in \mathbb{R}_{>0}^{n_i}$.

The most commonly studied class of games is zero-sum games. In a zero-sum game, there is a payoff matrix $A \in \mathbb{R}^{n_1 \times n_2}$ where player 1 receives utility $x_1 \cdot Ax_2$ and player 2 receives utility $-x_1 \cdot Ax_2$ resulting in the following optimization problem:

$$\max_{x_1 \in \mathcal{X}_1} \min_{x_2 \in \mathcal{X}_2} x_1 \cdot Ax_2 \qquad \text{(Two-Player Zero-Sum Game)}$$

The solution to this saddle problem is the Nash equilibrium $x^{NE}$. If player 1 selects her Nash equilibria $x_1^{NE}$, then she guarantees her utility is $x_1^{NE} \cdot Ax_2 \geq x_1^{NE} \cdot Ax_2^{NE}$ independent of what strategy player 2 selects. $x_1^{NE} \cdot Ax_2^{NE}$ is referred to as the value of the game.

### 2.1 Online Learning in Continuous Time

In many applications of game theory, players know neither the payoff matrix nor the Nash equilibria. In such settings, players select their strategies adaptively. The most common way to do this in continuous time is by using a follow-the-regularized-leader (FTRL) algorithm. Given a strongly convex regularizer, a learning rate $\eta$, and an initial payoff vector $y_i(0)$, players select their strategies at time $T$ according to

$$y_1(T) = y_1(0) + \int_0^T Ax_2(t)dt \qquad \text{(Player 1 Payoff Vector)}$$

$$y_2(T) = y_2(0) - \int_0^T A^\intercal x_1(t)dt \qquad \text{(Player 2 Payoff Vector)}$$

$$x_i(T) = \operatorname*{arg\,max}_{x_i \geq 0 : \sum_{j \in [n_i]} x_{ij} = 1} \left\{ y_i(T) \cdot x_i - \frac{h_i(x_i)}{\eta} \right\} \qquad \text{(Continuous FTRL)}$$

In this paper, we are primarily interested in the regularizer $h_i(x_i) = ||x_i||_2^2/2$ resulting in the Gradient Descent algorithm:

$$x_i(t) = \operatorname*{arg\,max}_{x_i \geq 0: \sum_{j \in [n_i]} x_{ij} = 1} \left\{ y_i(t) \cdot x_i - \frac{||x_i||_2^2}{2\eta} \right\} \qquad \text{(Continuous Gradient Descent)}$$

Continuous time FTRL learning in games has an interesting number of properties including time-average converge to the set of coarse correlated equilibria at a rate of $O(1/T)$ in general games (Mertikopoulos et al. [2018]) and thus to Nash equilibria in zero-sum games. These systems can also exhibit interesting recurrent behavior e.g. periodicity (Piliouras and Schulman [2018], Nagarajan et al. [2018]), Poincaré recurrence (Mertikopoulos et al. [2018], Piliouras and Shamma [2014], Piliouras et al. [2014]) and limit cycles (Kleinberg et al. [2011]). These systems have formal connections to Hamiltonian dynamics (i.e. energy preserving systems) (Bailey and Piliouras [2019]). All of these types of recurrent behavior are special cases of chain recurrence (Papadimitriou and Piliouras [2018], Omidshafiei et al. [2019]).

## 2.2   Online Learning in Discrete Time

In most settings, players update their strategies iteratively in discrete time steps. The most common class of online learning algorithms is again the family of follow-the-regularized-leader algorithms.

$$y_1^T = y_1^0 + \sum_{t=1}^{T-1} A x_2^t \qquad \text{(Player 1 Payoff Vector)}$$

$$y_2^T = y_2^0 - \sum_{t=1}^{T-1} A^\intercal x_1^t \qquad \text{(Player 2 Payoff Vector)}$$

$$x_i^t = \operatorname*{arg\,max}_{x_i \geq 0: \sum_{j \in [n_i]} x_{ij} = 1} \left\{ y_i^t \cdot x_i - \frac{h_i(x_i)}{\eta} \right\} \qquad \text{(FTRL)}$$

$$x_i^t = \operatorname*{arg\,max}_{x_i \geq 0: \sum_{j \in [n_i]} x_{ij} = 1} \left\{ y_i^t \cdot x_i - \frac{||x_i||_2^2}{2\eta} \right\} \qquad \text{(Gradient Descent)}$$

where $\eta$ corresponds to the learning rate. In Lemma 6 of Appendix B, we show (FTRL) is the first order approximation of (Continuous FTRL).

These algorithms again have interesting properties in zero-sum games. The time-average strategy converges to a $O(\eta)$-approximate Nash equilibrium (Cesa-Bianchi and Lugoisi [2006]). On the contrary, Bailey and Piliouras [2018] show that the day-to-day behavior diverges away from interior Nash equilibria. For notational simplicity we do not introduce different learning rates $\eta_1, \eta_2$ but all of our proofs immediately carry over to this setting.

## 2.3   Regret in Online Learning

The most common way of analyzing an online learning algorithm is by examining its regret. The regret at time/iteration $T$ is the difference between the accumulated utility gained by the algorithm and the total utility of the best fixed action with hindsight. Formally for player 1,

$$Regret_1(T) = \max_{x_1 \in \mathcal{X}_1} \left\{ \int_0^T x_1 \cdot Ax_2(t) dt \right\} - \int_0^T x_1(t) \cdot Ax_2(t) dt \qquad (1)$$

$$Regret_1(T) = \max_{x_1 \in \mathcal{X}_1} \left\{ \sum_{t=0}^T x_1 \cdot Ax_2^t \right\} - \sum_{t=0}^T x_1^t \cdot Ax_2^t \qquad (2)$$

for continuous and discrete time respectively.

In the case of (Continuous FTRL) it is possible to show rather strong regret guarantees. Specifically, Mertikopoulos et al. [2018] establish that $Regret_1(T) \in O(1)$ even for non-zero-sum games. In contrast, (FTRL) only guarantees $Regret_1(T) \in O(\eta \cdot T)$ for a fixed learning rate. In this paper, we utilize the geometry of (Gradient Descent) to show $Regret_1(T) \in O(\sqrt{T})$ in 2x2 zero-sum games ($n_1 = n_2 = 2$).

# 3 The Geometry of Gradient Descent

**Theorem 1.** *Let $A$ be a 2x2 game that has a unique fully mixed Nash equilibrium where strategies are updated according to (Gradient Descent). For any non-equilibrium initial strategies and any fixed learning rate $\eta$, there exists a $B$ such that $x^t$ is on the boundary for all $t \geq B$.*

Theorem 1 strengthens the result for (Gradient Descent) in 2x2 games from Bailey and Piliouras [2018]. Specifically, Bailey and Piliouras [2018] show that strategies come arbitrarily close to the boundary infinitely often when updated with any version of (FTRL). This is accomplished by closely studying the geometry of the player strategies. We strengthen this result for (Gradient Descent) in 2x2 games by focusing on the geometry of the payoff vectors. The proof of Theorem 1 relies on many of the tools developed in Section 4 for Theorem 3 and is deferred to Appendix G. The first step to understanding the trajectories of the dynamics of (Gradient Descent), is characterizing the solution to (Gradient Descent). The exact solution of (Gradient Descent) is described by Lemma 2 below.

**Lemma 2.** *The solution to (Gradient Descent) is given by*

$$x_{ij}^t = \begin{cases} 0 & for \ j \notin S_i \\ \eta \left( y_{ij}^t - \sum_{k \in S_i} \frac{y_{ik}^t}{|S_i|} \right) + \frac{1}{|S_i|} & for \ j \in S_i \end{cases}. \tag{3}$$

*where $S_i$ is found using Algorithm 1.*

---
**Algorithm 1** Finding Optimal Set $S_i$
---
1: **procedure** FIND $S_i$
2:      $S_i \leftarrow [n_i]$
3: *Search:*
4:      Select $j \in \arg\min_{k \in S_i} \{y_{ik}^t\}$
5:      **if** $\eta \left( y_{ij}^t - \sum_{k \in S_i} \frac{y_{ik}^t}{|S_i|} \right) + \frac{1}{|S_i|} < 0$
6:          $S_i \leftarrow S_i \setminus \{j\}$
7:          **goto** *Search*
8:      **else**
9:          **return** $S_i$
---

We defer the proof of Lemma 2 to Appendix C.

## 3.1 Convex Conjugate of the Regularizer

Our analysis primarily takes place in the space of payoff vectors. The payoff vector $y_i^t$ is a formal dual of the strategy $x_i^t$ obtained via

$$h^*(y_i^t) = \max_{x_i \geq 0 : \sum_{j \in [n_i]} x_{ij} = 1} \left\{ y_i^t \cdot x_i - \frac{h_i(x_i)}{\eta} \right\} \tag{4}$$

which is known as the convex conjugate or Fenchel Coupling of $h_i$ and is closely related to the Bregman Divergence. Mertikopoulos et al. [2018] and Bailey and Piliouras [2019] show that the "energy" $r = \sum_{i=1}^{2} h_i^*(y_i^t)$ is conserved in (Continuous FTRL). By Lemma 6, (FTRL) is the first order approximation of (Continuous FTRL). The energy $\{y : r \leq \sum_{i=1}^{2} h_i^*(y_i)\}$ is convex, and therefore the energy will be non-decreasing in (FTRL). Bailey and Piliouras [2018] capitalized on this non-decreasing energy to show that strategies come arbitrarily close to the boundary infinitely often in (FTRL).

In a similar fashion, we precisely compute $h^*(y_i^t)$ to better understand the dynamics of (Gradient Descent). We deviate slightly from traditional analysis of (FTRL) and embed the learning rate $\eta$ into the regularizer $h_i(x_i^t)$. Formally, define $h_i(x_i^t) = ||x_i^t||_2^2/(2\eta)$. Through the maximizing argument (Kakade et al. [2009]), we have

$$h_i^*(y_i^t) = y_i^t \cdot x_i^t - \frac{||x_i^t||_2^2}{2\eta}. \tag{5}$$

From Lemma 2,

$$h_i^*(y_i^t) = y_i^t \cdot x_i^t - \frac{||x_i^t||_2^2}{2\eta} = \frac{\eta}{2}\sum_{j \in S_i}(y_{ij}^t)^2 + \sum_{j \in S_i}\frac{y_{ij}^t}{|S_i|} - \frac{\eta}{2}\frac{\left(\sum_{j \in S_i}y_{ij}^t\right)^2}{|S_i|} - \frac{1}{2\eta}\frac{1}{|S_i|}. \qquad (6)$$

## 3.2 Selecting the Right Dual Space in 2x2 Games

Since $h_i(x_i) = ||x_i||_2^2/(2\eta)$ is a strongly smooth function in the simplex, we expect for $h_i^*(y_i)$ to be strongly convex (Kakade et al. [2009]) – at least when it's corresponding dual variable $x_i$ is positive. However, (6) is not strongly convex for all $y_i^t \in \mathbb{R}^{n_i}$. This is because $y_i^{t+1}$ cannot appear anywhere in $\mathbb{R}^{n_i}$. Rather, $y_i^{t+1}$ is contained to a space $\mathcal{X}_i^*$ dual to the domain $\{x_i \in \mathbb{R}_{\geq 0}^{n_i} : \sum_{j=1}^{n_i}x_{ij} = 1\}$.

There are many non-intersecting dual spaces for the payoff vectors that yield strategies $\{x_i^t\}_{t=1}^{\infty}$. Mertikopoulos et al. [2018] informally define a dual space when they focus the analysis on the vector $y_i(t) - y_{in_i}(t)\mathbf{1}$. Similarly, we define a dual space that will be convenient for showing our results in 2x2 zero-sum games. Consider the payoff matrix

$$A = \begin{bmatrix} a & b \\ c & d \end{bmatrix} \qquad (7)$$

Without loss of generality, we may assume $a > \min\{0, b, c\}$, $d > \min\{0, b, c\}$, and $A$ is singular, i.e., $ad - bc = 0$ (see Appendix D for details). Denote $\Delta y_1^t$ as

$$\Delta y_1^t = y_1^{t+1} - y_1^t \qquad (8)$$
$$= Ax_2^t \qquad (9)$$
$$= \begin{bmatrix} (a-b)x_{21}^t + b \\ (c-d)x_{21}^t + d \end{bmatrix} \qquad (10)$$

Therefore

$$[d-c, a-b] \cdot \Delta y_1^t = ad - bc = 0 \qquad (11)$$

since $A$ is singular. When $y_{11}^t$ increases by $a-b$, $y_{12}^t$ increases by $c-d$. Thus, the vector $[a-b, c-d]$ describes the span of the dual space $\mathcal{X}_1^*$. Moreover, (FTRL) is invariant to constant shifts in the payoff vector $y_1^t$ and therefore we may assume $[d-c, a-b] \cdot y_1^0 = 0$. By induction,

$$[d-c, a-b] \cdot y_1^t = [d-c, a-b] \cdot (y_1^{t-1} + \Delta y_1^{t-1}) \qquad (12)$$
$$= [d-c, a-b] \cdot y_1^{t-1} = 0 \qquad (13)$$

This conveniently allows us to express $y_{12}^t$ in terms of $y_{11}^t$,

$$y_{12}^t = \frac{c-d}{a-b}y_{11}^t. \qquad (14)$$

Symmetrically,

$$y_{22}^t = \frac{b-d}{a-c}y_{21}^t. \qquad (15)$$

Combining these relationships with Lemma 2 yields

$$x_{11}^t = \begin{cases} 0 & \text{if } \eta\left(1 - \frac{c-d}{a-b}\right)\frac{y_{11}^t}{2} + \frac{1}{2} \leq 0 \\ 1 & \text{if } \eta\left(1 - \frac{c-d}{a-b}\right)\frac{y_{11}^t}{2} + \frac{1}{2} \geq 1 \\ \eta\left(1 - \frac{c-d}{a-b}\right)\frac{y_{11}^t}{2} + \frac{1}{2} & \text{otherwise} \end{cases} \qquad (16)$$

$$x_{21}^t = \begin{cases} 0 & \text{if } \eta\left(1 - \frac{b-d}{a-c}\right)\frac{y_{21}^t}{2} + \frac{1}{2} \leq 0 \\ 1 & \text{if } \eta\left(1 - \frac{b-d}{a-c}\right)\frac{y_{21}^t}{2} + \frac{1}{2} \geq 1 \\ \eta\left(1 - \frac{b-d}{a-c}\right)\frac{y_{21}^t}{2} + \frac{1}{2} & \text{otherwise} \end{cases} \qquad (17)$$

The selection of this dual space also allows us to employ a convenient variable substitution to plot $x^t$ and $y^t$ on the same graph.

$$z_1^t = \eta \left( 1 - \frac{c-d}{a-b} \right) \frac{y_{11}^t}{2} + \frac{1}{2} \tag{18}$$

$$z_2^t = \eta \left( 1 - \frac{b-d}{a-c} \right) \frac{y_{21}^t}{2} + \frac{1}{2} \tag{19}$$

The strategy $x^t$ can now be expressed as

$$x_{i1}^t = \begin{cases} 0 & \text{if } z_i^t \leq 0 \\ 1 & \text{if } z_i^t \geq 1 \\ z_i^t & \text{otherwise} \end{cases} \tag{20}$$

Moreover, (6) can be rewritten as

$$h_i^*(y_1^t) = \bar{h}_1^*(z_1^t) = \begin{cases} \alpha_{10} z_1^t - \beta_{10} & \text{if } z_1^t \leq 0 \\ \alpha_{11} z_1^t - \beta_{11} & \text{if } z_1^t \geq 1 \\ \gamma_1 (z_1^t)^2 + \alpha_1 z_1^t - \beta_1 & \text{otherwise} \end{cases} \tag{21}$$

$$h_i^*(y_2^t) = \bar{h}_2^*(z_2^t) = \begin{cases} \alpha_{20} z_2^t - \beta_{20} & \text{if } z_2^t \leq 0 \\ \alpha_{21} z_2^t - \beta_{21} & \text{if } z_2^t \geq 1 \\ \gamma_2 (z_2^t)^2 + \alpha_2 z_2^t - \beta_2 & \text{otherwise} \end{cases} \tag{22}$$

where $\alpha_{i0} < 0, \alpha_{i1} > 0$, and $\gamma_i > 0$. Both of these expressions are obviously strongly convex when the corresponding player strategy is in $(0, 1)$. The full details of these reduction can be found in Appendix E. With this notation, $(x_{11}^t, x_{21}^t)$ is simply the projection of $z^t$ onto the unit square as shown in Figure 2.

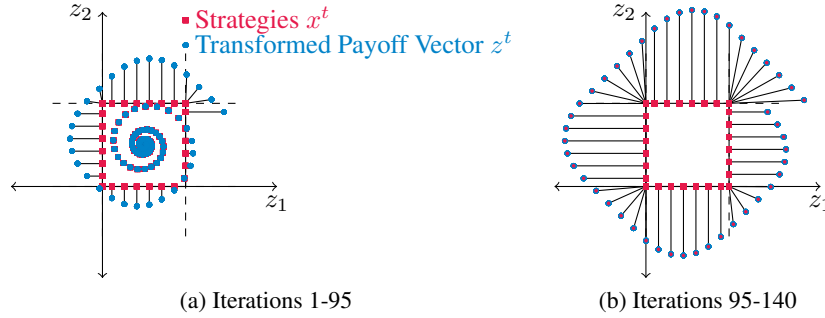

(a) Iterations 1-95     (b) Iterations 95-140

Figure 2: Strategies and Transformed Payoff Vectors Rotating Clockwise and Outwards in Matching Pennies with $\eta = .15$ and $(y_{11}^0, y_{11}^0) = (.2, -.3)$.

## 4  $\Theta(\sqrt{T})$ Regret in 2x2 Zero-Sum Games

**Theorem 3.** *Let $A$ be a 2x2 game that has a unique fully mixed Nash equilibrium. When $x^t$ is updated according to (Gradient Descent) with any fixed learning rate $\eta$, $Regret_1(T) \in O\left(\sqrt{T}\right)$.*

It is well known that if an algorithm admits sublinear regret in zero-sum games, then the time-average play converges to a Nash equilibirum. Thus, Theorem 3 immediately results in the following corollary.

**Corollary 4.** *Let $A$ be a 2x2 game that has a unique fully mixed Nash equilibrium. When $x^t$ is updated according to (Gradient Descent) with any fixed learning rate $\eta$, the average strategy $\bar{x}^T = \sum_{t=1}^T \frac{x^t}{T}$ converges to $x^{NE}$ as $T \to \infty$.*

*Proof of Theorem 3.* The result is simple if $x^1 = x^{NE}$. Neither player strategy will ever change. Since player 1's opponent is playing the fully mixed $x_2^{NE}$, player 1's utility is constant independent of what strategy is selected and therefore the regret is always 0. Now consider $x^1 \neq x^{NE}$.

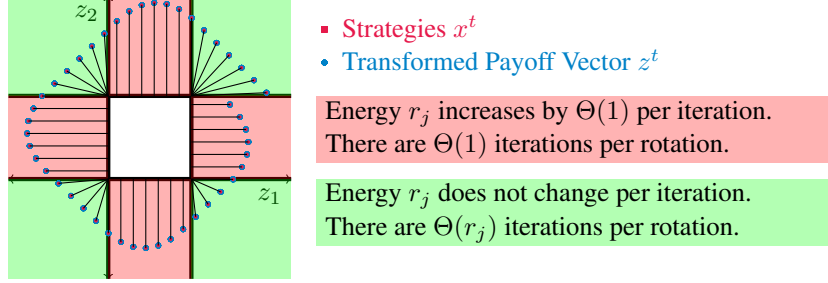

Figure 3: Partitioning of Payoff Vectors for the Proof of Theorem 3.

The main details of the proof are captured in Figure 3. Specifically in Appendix F.1, we establish break points $t_0 < t_1 < ... < t_k = T + 1$ and analyze the impact strategies $x^{t_j}, x^{t_j+1}, ..., x^{t_{j+1}-1}$ have on the regret. The strategies $x^{t_j}, x^{t_j+1}, ..., x^{t_{j+1}-1}$ are contained in adjacent red and green sections as shown in Figure 3.

In Appendix F.2, we show that there exists $\Theta(1)$ iterations where $x^t \neq x^{t+1}$ for each partitioning, $\{t_j, t_j + 1, ..., t_{j+1} - 1\}$. Specifically, we show that $\Theta(1)$ consecutive payoff vectors appear in a red section of Figure 3. The remaining points all appear in a green section and the corresponding player strategies are equivalent. This implies

$$\sum_{t=t_j}^{t_{j+1}-1} (x_1^{t+1} - x_1^t) \cdot Ax_2^t = \sum_{t \in [t_j, t_{j+1}-1]: x_1^{t+1} \neq x_1^t} (x_1^{t+1} - x_1^t) \cdot Ax_2^t \tag{23}$$

$$\in \sum_{t \in [t_j, t_{j+1}-1]: x_1^{t+1} \neq x_1^t} O(1) \in O(1) \tag{24}$$

Denote $r_j = \sum_{i=1}^{2} \bar{h}_i^*(z_i^{t_j})$ as the total energy of the system in iteration $t_j$. In Appendix F.3, we show this energy increases linearly in each partition, i.e., $r_{j+1} - r_j \in \Theta(1)$. In Appendix F.4, we also show that the size of each partition is proportional to the energy in the system at the beginning of that partition, i.e., $t_{j+1} - t_j \in \Theta(r_j)$. Combining these two, $t_j \in \Theta(j^2)$. Therefore $T \in \Theta(k^2)$ and $k \in \Theta\left(\sqrt{T}\right)$ where $k$ is the total number of partitions. Finally, it is well known (Cesa-Bianchi and Lugoisi [2006]) that the regret of player 1 in zero-sum games through $T$ iterations is bounded by

$$Regret_1(T) \leq O(1) + \sum_{t=0}^{T} (x_1^{t+1} - x_1^t) \cdot Ax_2^t \tag{25}$$

$$\leq O(1) + \sum_{t=0}^{t_0-1} (x_1^{t+1} - x_1^t) \cdot Ax_2^t + \sum_{i=1}^{k} \sum_{t=t_{i-1}}^{t_i-1} (x_1^{t+1} - x_1^t) \cdot Ax_2^t \tag{26}$$

$$\in O(1) + \sum_{i=1}^{k} O(1) \in O\left(\sqrt{T}\right) \tag{27}$$

completing the proof of the theorem. $\qquad\square$

Next, we provide a game and initial conditions that has regret $\Theta(\sqrt{T})$ establishing that the bound in Theorem 3 is tight.

**Theorem 5.** *Consider the game Matching Pennies with fixed learning rate $\eta = 1$ and initial conditions $y_1^0 = y_2^0 = (1, 0)$. Then player 1's regret is $\Theta(\sqrt{T})$ when strategies are updated with (Gradient Descent).*

The proof follows similarly to the proof of Theorem 3 by exactly computing the regret in every iteration of (Gradient Descent). The full details appear in Appendix H.

# 5   Higher Dimensions and Other Regularizers

Many of the techniques introduced in this paper extend both to higher dimensions for Gradient Descent and to other variants of FTRL. Our proof consists mainly of three parts:

1. the "step-size" in the dual space is bounded; i.e., $||y_i^t - y_i^{t-1}|| \leq b$ for some constant $b$.
2. a proof of divergence in the dual space where the divergence grows linearly when at least one agent is not playing a pure strategy and negligibly when both agents are playing a pure strategy.
3. a proof of recurrence where the "cycle" length (in the primal/strategy space) is bounded

The first two components immediately extend to higher dimensions using the current analysis. In regards to the last step, recent advancements in understanding the geometry of learning dynamics in larger games (e.g., Mertikopoulos et al. [2018], Bailey and Piliouras [2019]) suggest that, although non-trivial, this last step can also be eventually rigorously established. However, new ideas are most likely needed to for the last step. In Appendix I, we provide more evidence for sublinear regret in higher dimensions including experiments suggesting that regret grows at approximately $O(\sqrt{T})$ even when the number of strategies is large.

It is also likely that sublinear regret extends to other variants of FTRL using a similar analysis. In two-by-two zero-sum games, both steps (1) and (3) trivially extend for other variants of FTRL. As we discuss further in Appendix I, the proof for (2) relies primarily on the strict convexity of the regularizer $h$ – a property shared by all variants of FTRL. For Gradient Descent, we make use of this property by showing divergence increases as the strategies move from one pure strategy to another.

However, strategies will never reach the boundary for some variants of FTRL. For example, the multiplicative weights update algorithm always selects fully-mixed strategies and (2) does not hold exactly as written. Instead, for any $\epsilon$, after a finite number of iterations all strategies will appear within $\epsilon$ of the boundary. This proof follows identically to the proof for Gradient Descent in Appendix G. Moreover, the first part of (2) extends to this settings; when one agent is more than $\epsilon$ away from the boundary, the divergence grows linearly. However, to prove that the divergence grows negligibly when both players are within $\epsilon$ of the boundary, we will have to carefully evolve $\epsilon$ over time. This is because for an algorithm like multiplicative weights, the convex conjugate $h^*$ is never linear; rather it becomes arbitrarily close to a linear function as both agents come closer to playing a pure strategy. Alternatively, (2) will more readily follow upon establishing a tighter understanding of the geometry of learning dynamics.

For both higher dimensions and other variants of FTRL, this work provides evidence that regret grows sublinearly when both agents are using fixed step-size. More importantly, it establishes an outline on the proof that relies on further developments in understanding the trajectories of online learning dynamics.

# 6   Conclusion

We present the first proof of sublinear regret for the most classic FTRL dynamic, online gradient descent, in two-by-two zero-sum games. Our proof techniques leverage geometric information and hinge upon the fact that FTRL dynamics, although are typically referred to as "converging" to Nash equilibria in zero-sum games, diverge away from them. Our simulations further suggest that sublinear regret bounds carry over to larger zero-sum games.

# 7   Acknowledgements

James P. Bailey and Georgios Piliouras acknowledge MOE AcRF Tier 2 Grant 2016-T2-1-170, grant PIE-SGP-AI-2018-01 and NRF 2018 Fellowship NRF-NRFF2018-07.

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
