[Supplementary Material]

# Fast and Furious Learning in Zero-Sum Games: Vanishing Regret with Non-Vanishing Step Sizes

## Supplementary Materials

**James P. Bailey**
Texas A&M University
jamespbailey@tamu.edu

**Georgios Piliouras**
Singapore University of Technology and Design
georgios@sutd.edu.sg

## A   Other Related Work

*Non-equilibration and chaos in game dynamics.* In recent years the algorithmic game theory community has produced several interesting non-equilibrium results. Palaiopanos et al. [2017], Chotibut et al. [2018] prove the existence of Li-Yorke chaos for multiplicative weights update (MWU) in 2x2 potential games. Chaos also emerges for MWU in non-atomic linear congestion games with just two paths Chotibut et al. [2019]. For the case of zero-sum games, Bailey and Piliouras [2018] showed that Nash equilibria are repelling for all follow-the-regularized-leader (FTRL) dynamics (even for 2x2 games such as Matching Pennies). Cheung and Piliouras [2019] proved that FTRL dynamics are furthermore chaotic in zero-sum games. Pangallo et al. [2017] established experimentally that in 2x2 games between agents of opposing interests a large class of dynamics typically result in limit cycles and chaos. Our result adds a new chapter in this area with new detailed understanding of the non-equilibrium trajectories of gradient descent in two-by-two zero-sum games and their implications to regret.

*Fast regret minimization in games.* It is well known that MWU can achieve (time-average) regret of $O(1/\sqrt{t})$ by using step size of $(1/\sqrt{t})$ without making any assumptions about its environment. Daskalakis et al. [2011] and Rakhlin and Sridharan [2013] developed no-regret dynamics with a $O(\log t/t)$ regret minimization rate when played against each other in zero-sum games. Syrgkanis et al. [2015] analyzed a recency biased variant of FTRL and showed regret of $O(t^{-3/4})$ in general games. The social welfare converges at a rate of $O(t^{-1})$. This was extended to standard versions of FRTL dynamics by Foster et al. [2016].

*Learning in zero-sum games and Machine Learning applications.* A stream of recent papers proves positive results about convergence to equilibria in (mostly bilinear) zero-sum games for suitable adapted variants of first-order methods and then apply these techniques to Generative Adversarial Networks (GANs), showing improved performance (e.g. Daskalakis et al. [2018], Balduzzi et al. [2018], Mertikopoulos et al. [2019], Daskalakis and Panageas [2019], Gidel et al. [2019], Yazıcı et al. [2019]) FTRL dynamics in continuous-time exhibit conservation laws and recurrent cycle-like behavior in zero-sum games and have formal connections to Hamiltonian systems (Piliouras and Shamma [2014], Mertikopoulos et al. [2018], Bailey and Piliouras [2019]). Exploiting these connections novel discretization schemes can be developed with finite regret in general games (Bailey et al. [2019]).

Finally, the emergence of cycles in games lies at the core of some of the most exciting problems in creating artificial agents for complex environments such as Starcraft, where even evaluating the strength of an individual agent is a non-trivial task (Balduzzi et al. [2018], Omidshafiei et al. [2019]). Recent approaches are inspired by the emergence of cyclic behavior to introduce algorithms that aim at game-theoretic niching (Balduzzi et al. [2019]).

## B First Order Approximation of (Continuous FTRL)

**Lemma 6.** *(FTRL) is the first order approximation of (Continuous FTRL).*

*Proof.* The first order approximation of $y_1(t)$ is

$$\hat{y}_1(t) = y_1(t-1) + \frac{d}{dt}y_1(t-1) \tag{28}$$

$$= y_1(t-1) + Ax_1(t-1) \tag{29}$$

and

$$\hat{x}_1(t) = \underset{x_1 \in \mathcal{X}_1}{\arg\max} \left\{ x \cdot \hat{y}_1(t) - \frac{h_1(x_1)}{\eta} \right\} \tag{30}$$

Inductively, $\hat{y}_1(t) = y_1^t$ and $\hat{x}_1(t) = x_1^t$ as defined in (FTRL) completing the proof of the lemma. $\square$

## C Proof of Lemma 2

The KKT optimality conditions (see Bertsekas [1999]) for (Gradient Descent) are given by

$$x_i^t = \eta\left(y_i^t - \lambda_i^t \cdot \mathbf{1} + u_i^t\right) \qquad \text{(Critical Point)}$$

$$x_i^t \geq 0 \qquad \text{(Non-negativity)}$$

$$\sum_{j=1}^{n_i} x_{ij}^t = 1 \qquad \text{(Primal Feasibility)}$$

$$u_i^t \geq 0 \qquad \text{(Dual Feasibility)}$$

$$u_i^t \cdot x_i^t = 0 \qquad \text{(Complimentary Slackness)}$$

where $u_i^t \in \mathbb{R}^{n_i}$ and $\lambda_i^t \in \mathbb{R}$.

Let $S_i$ be the set of $j$ where $u_{ij}^t = 0$. By (Complimentary Slackness), $x_{ij}^t = 0$ for all $j \notin S_i$. Therefore, (Critical Point) becomes

$$x_{ij}^t = \begin{cases} 0 & \text{for } j \notin S_i \\ \eta(y_{ij}^t - \lambda_i^t) & \text{for } j \in S_i \end{cases}. \tag{31}$$

Substituting (31) into (Primal Feasibility) yields

$$1 = \sum_{j=1}^{n_i} x_{ij}^t \tag{32}$$

$$= \sum_{j \in S_i} \eta(y_{ij}^t - \lambda_i^t) \tag{33}$$

and $\lambda_i^t = \sum_{j \in S_i} y_{ij}^t / |S_i| - 1/(\eta|S_i|)$. Therefore

$$x_{ij}^t = \begin{cases} 0 & \text{for } j \notin S_i \\ \eta\left(y_{ij}^t - \sum_{k \in S_i} \frac{y_{ik}^t}{|S_i|}\right) + \frac{1}{|S_i|} & \text{for } j \in S_i \end{cases}. \tag{34}$$

The variable $u_{ij}^t = 0$ represents that the constraint $x_{ij}^t$ is unenforced. Enforcing constraints never improves the objective value of an optimization problem and therefore $S_i \subseteq [n_i]$ is a maximal set where (34) is feasible. Moreover, it is straightforward to show that if $y_{ij}^t \geq y_{ik}^t$ then $x_{ij}^t \geq x_{ik}^t$. Thus, greedily removing the lowest valued $y_{ij}^t$ from $\hat{S}_i = [n_i]$ until (34) is feasible yields the optimal solution to (Gradient Descent).

# D  Payoff Matrix Assumptions

The payoff matrix is in the form

$$A = \begin{bmatrix} a & b \\ c & d \end{bmatrix} \tag{35}$$

In this paper, we make three assumptions about $A$: $ad - bc = 0$, $a > \max\{0, b, c\}$ and $d > \max\{0, b, c\}$. In order, we show that we may make these assumption without loss of generality.

In 2x2 games, if there is a unique fully mixed Nash equilibrium, then it is straight forward to show that player 2's equilibrium is

$$x_2^{NE} = \left( \frac{d - b}{a + d - b - c}, \frac{a - c}{a + d - b - c} \right) \tag{36}$$

and therefore $a + d - b - c \neq 0$, $d \neq b$ and $a \neq c$ when there is a unique fully mixed Nash equilibrium. Similarly, by analyzing player 1's Nash equilibrium, $d \neq c$ and $a \neq b$. Now consider the payoff matrix

$$B = \begin{bmatrix} a + \frac{ad-bc}{a+d-b-c} & b + \frac{ad-bc}{a+d-b-c} \\ c + \frac{ad-bc}{a+d-b-c} & d + \frac{ad-bc}{a+d-b-c} \end{bmatrix} \tag{37}$$

The determinant of payoff matrix $B$ is zero. Moreover, (FTRL) is invariant to shifts in the payoff matrix, so for the purpose of the dynamics $\{x^t\}_{t=1}^{\infty}$, $A$ and $B$ are equivalent matrices. Thus, without loss of generality we may assume the payoff matrix is singular by shifting the matrix by a specific constant.

Next, we argue that we may assume $a > 0$. Players 1 and 2 separately try to solve

$$\max_{x_1 \in \mathcal{X}_1} \min_{x_2 \in \mathcal{X}_2} x_1 \cdot \begin{bmatrix} a & b \\ c & d \end{bmatrix} x_2 = \max_{x_1 \in \mathcal{X}_1} \min_{x_2 \in \mathcal{X}_2} -x_1 \cdot \begin{bmatrix} -a & -b \\ -c & -d \end{bmatrix} x_2 \tag{38}$$

$$= -\max_{x_2 \in \mathcal{X}_2} \min_{x_1 \in \mathcal{X}_1} x_2 \cdot \begin{bmatrix} -a & -c \\ -b & -d \end{bmatrix} x_1. \tag{39}$$

Thus, by possibly switching the maximization and minimization roles between player 1 and player 2, we may assume $a > 0$.

Next we show that we may assume $a > \max\{b, c\}$. If $a + d - b - c > 0$ then (36) implies $a > c$ and, symmetrically, $a > b$ completing the claim. If instead, $a + d - b - c < 0$, then through identical reasoning, $\min\{b, c\} > a > 0$ and we can simply rewrite the payoff matrix as

$$\max_{x_1 \in \mathcal{X}_1} \min_{x_2 \in \mathcal{X}_2} x_1 \cdot \begin{bmatrix} a & b \\ c & d \end{bmatrix} x_2 = \max_{x_1 \in \mathcal{X}_1} \min_{x_2 \in \mathcal{X}_2} x_1 \cdot \begin{bmatrix} b & a \\ d & c \end{bmatrix} x_2 \tag{40}$$

With the new payoff matrix, $b + c - a - d > 0$ implying $b > \max\{a, d\} \geq 0$ as desired. Thus, we may assume $a > \max\{0, b, c\}$ by relabeling player 1's strategies.

Finally, $ad - bc = 0$ and $a > \max\{0, b, c\}$ implies $d > \max\{0, b, c\}$. The prior analysis argues $a + d - b - c > 0$. Thus, (36) implies $d > \max\{b, c\}$. Now for contradiction, suppose $d < 0$. This implies $0 > d > \max\{b, c\}$ and $ad - bc < 0$ a contradiction. Therefore $d > \max\{0, b, c\}$.

# E  Expressing the Convex Conjugate with the Transformed Payoffs

We can express $x$ as

$$x_{11}^t = \begin{cases} 0 & \text{if } \eta \left( 1 - \frac{c-d}{a-b} \right) \frac{y_{11}^t}{2} + \frac{1}{2} \leq 0 \\ 1 & \text{if } \eta \left( 1 - \frac{c-d}{a-b} \right) \frac{y_{11}^t}{2} + \frac{1}{2} \geq 1 \\ \eta \left( 1 - \frac{c-d}{a-b} \right) \frac{y_{11}^t}{2} + \frac{1}{2} & \text{otherwise} \end{cases} \tag{41}$$

$$x_{21}^t = \begin{cases} 0 & \text{if } \eta \left( 1 - \frac{b-d}{a-c} \right) \frac{y_{21}^t}{2} + \frac{1}{2} \leq 0 \\ 1 & \text{if } \eta \left( 1 - \frac{b-d}{a-c} \right) \frac{y_{21}^t}{2} + \frac{1}{2} \geq 1 \\ \eta \left( 1 - \frac{b-d}{a-c} \right) \frac{y_{21}^t}{2} + \frac{1}{2} & \text{otherwise.} \end{cases} \tag{42}$$

Thus, (6) simplifies to

$$
h_1^*(y_1^t) = \begin{cases} y_{12}^t - \frac{1}{2\eta} & \text{if } x_{11}^t = 0 \\ y_{11}^t - \frac{1}{2\eta} & \text{if } x_{11}^t = 1 \\ \frac{\eta}{4}\left(y_{11}^t - y_{12}^t\right)^2 + \frac{y_{11}^t + y_{12}^t}{\eta} - \frac{1}{4\eta} & \text{otherwise} \end{cases} \tag{43}
$$

$$
= \begin{cases} \frac{c-d}{a-b} y_{11}^t - \frac{1}{2\eta} & \text{if } x_{11}^t = 0 \\ y_{11}^t - \frac{1}{2\eta} & \text{if } x_{11}^t = 1 \\ \frac{\eta}{4}\left(1 - \frac{c-d}{a-b}\right)^2 (y_{11}^t)^2 + \frac{\left(1 - \frac{c-d}{a-b}\right) y_{11}^t}{\eta} - \frac{1}{4\eta} & \text{otherwise} \end{cases} \tag{44}
$$

Symmetrically,

$$
h_2^*(y_2^t) = \begin{cases} \frac{b-d}{a-c} y_{21}^t - \frac{1}{2\eta} & \text{if } x_{21}^t = 0 \\ y_{21}^t - \frac{1}{2\eta} & \text{if } x_{21}^t = 1 \\ \frac{\eta}{4}\left(1 - \frac{b-d}{a-c}\right)^2 (y_{21}^t)^2 + \frac{\left(1 - \frac{b-d}{a-c}\right) y_{21}^t}{\eta} - \frac{1}{4\eta} & \text{otherwise} \end{cases} \tag{45}
$$

Unlike (6), we can easily verify $h^*$ is strongly convex when the strategy is fully mixed. In addition to allowing for a simpler analysis,

$$
h_1^*(y_1^t) = \begin{cases} \frac{c-d}{a-b} y_{11}^t - \frac{1}{2\eta} & \text{if } x_{11}^t = 0 \\ y_{11}^t - \frac{1}{2\eta} & \text{if } x_{11}^t = 1 \\ \frac{\eta}{4}\left(1 - \frac{c-d}{a-b}\right)^2 (y_{11}^t)^2 + \frac{\left(1 - \frac{c-d}{a-b}\right) y_{11}^t}{\eta} - \frac{1}{4\eta} & \text{otherwise} \end{cases} \tag{46}
$$

$$
= \begin{cases} \alpha_{10} z_1^t - \beta_{10} & \text{if } z_1^t \le 0 \\ \alpha_{11} z_1^t - \beta_{11} & \text{if } z_1^t \ge 1 \\ \gamma_1 (z_1^t)^2 + \alpha_1 z_1^t - \beta_1 & \text{otherwise} \end{cases} \tag{47}
$$

$$
= \bar{h}_1^*(z_1^t) \tag{48}
$$

where $\alpha_{10} < 0, \alpha_{11} > 0$, and $\gamma_1 > 0$. Symmetrically,

$$
\bar{h}_2^*(z_2^t) = \begin{cases} \alpha_{20} z_2^t - \beta_{20} & \text{if } z_2^t \le 0 \\ \alpha_{21} z_2^t - \beta_{21} & \text{if } z_2^t \ge 1 \\ \gamma_2 (z_2^t)^2 + \alpha_2 z_2^t - \beta_2 & \text{otherwise} \end{cases} \tag{49}
$$

with $\alpha_{20} < 0, \alpha_{21} > 0$, and $\gamma_2 > 0$.

## F    Details of Theorem 3

### F.1    Partitioning the Strategies and the Dual Space

By assumption $a > \min\{0, b, c\}$ and $d > \min\{0, b, c\}$ (See Appendix D). This implies that both the strategies $(x_{11}, x_{21})$ and the transformed payoff vector $z$ will rotate clockwise about the Nash equilibrium in both continuous and discrete time as depicted in Figure 2. To formally show clockwise movement, assume $x_{11}^t \ge x_{11}^{NE}, x_{21}^t \ge x_{21}^{NE}$ (upper right of the Nash equilibrium). Then $x_1^t \cdot Ax_2^t \le (1,0) \cdot Ax_2^t$ implying $x_{11}^t \le x_{11}^{t+1}$. Symmetrically, $x_{21}^t \ge x_{22}^{t+1}$ implying that if $x_{11}^t \ge x_{11}^{NE}$ and $x_{21}^t \ge x_{21}^{NE}$ then the strategies move clockwise or not at all. Similarly, clockwise movement can be shown for the other three cases. A symmetric argument shows the transformed payoff vector $z$ also rotates clockwise.

To partition the strategies $\{x^t\}_{t=1}^T$, we begin by first partitioning the dual space $X^*$ into 4 regions $Z_0, Z_1, Z_2$, and $Z_3$. The visual representation of this partitioning is given in Figure 4.

$$
Z_0 = \{z : z_1 < 1, z_2 \ge 1\}.
$$
$$
Z_1 = \{z : z_1 \ge 1, z_2 > 0\}.
$$
$$
Z_2 = \{z : z_1 > 0, z_2 \le 0\}.
$$
$$
Z_3 = \{z : z_1 \le 0, z_2 < 1\}.
$$

The partitioning $Z_0$, $Z_1$, $Z_2$, and $Z_3$, is not a proper partitioning. As depicted in Figure 4, it lacks all payoff vectors that correspond to fully mixed strategies for both players. However, by Theorem 1, there exists a $B$ so that $x^t$ is not fully mixed for both players for all $t \geq B$. Since $B$ is finite, the first $B$ strategies will shift the total regret by at most a constant and therefore can be disregarded in our analysis.

Figure 4: Visual Representation of $Z_0, Z_1, Z_2$ and $Z_3$.

Since strategies move clockwise, in general the payoff vectors will move from region $Z_i$ to region $Z_{(i+1 \mod 4)}$. If $\eta$ is large, then it is possible to move directly from $Z_i$ to $Z_{(i+2 \mod 4)}$. While we consider such $\eta$ impractical, our analysis handles such cases and shows that after enough iterations, the payoff vectors never skip a region. Finally, we are able to define our partitioning over $\{x_t\}_{t=1}^T$. Let $B$ be as in the statement of Theorem 1 and let $Z(t) \in \{Z_0, Z_1, Z_2, Z_3\}$ be such that $z^t \in Z(t)$.

$$t_0 = \underset{t \geq B}{\arg\min}\{z^t \in Z_0\} \tag{50}$$

$$t_j = \underset{t \geq t_{j-1}}{\arg\min}\{z^t \notin Z(t-1)\} \; \forall j = 1, 2, \dots \tag{51}$$

Finally, let $t_k = T + 1$ where $k - 1$ is the largest index that has a solution in (51). The value $t_j$ represents the first time after $t_{j-1}$ that $z^t$ enters a new region. Our analysis now focuses on the time intervals created by these break points. Specifically, we analyze $x^{t_j}, x^{t_j+1}, \dots, x^{t_{j+1}-1}$ and $z^{t_j}, z^{t_j+1}, \dots, z^{t_{j+1}-1}$

## F.2 Player Strategies Often Do Not Change

In this section, we show that for each partitioning $\{t_j, \dots, t_{j+1} - 1\}$ the strategies change at most a constant, $\kappa$, of times independent of the size of the partitioning, $t_{j+1} - t_j$. This result is useful in two areas. First, in the proof of Theorem 3 it is used to show that $x^{t_j}, \dots, x^{t_{j+1}-1}$ contributes to the regret by an amount proportional to $\kappa$. Second, it is used in the proof of Lemma 8 to show the total energy in the system increases by a constant in each partition; we show the energy only increases when the player strategies change and therefore, the energy increases at most $\kappa$ times in each partition.

**Lemma 7.** *There exists a $\kappa$ such that $|\{t \in \{t_j, \dots, t_{j+1} - 1\} : x^t \neq x^{t+1}\}| \leq \kappa$ for all $j$.*

*Proof of Lemma 7.* Without loss of generality, assume $z^{t_j}, \dots, z^{t_{j+1}-1} \in Z_1$. This implies $x_{11}^{t_j} = \dots = x_{11}^{t_{j+1}-1} = 1$ and therefore

$$y_{21}^{t+1} - y_{21}^t = [-a, -c] \cdot [x_{11}^t, 1 - x_{11}^t] \tag{52}$$

$$= -a \tag{53}$$

for all $t = t_j, \dots, t_{j+1} - 1$. Thus, there must exist a constant $\delta_1 > 0$ such that $z_2^t - z_2^{t+1} = \delta_1$.

By selection of $Z_1$, $z_2^t > 0$ for all $t$. Moreover, $x_{21}^t = 1$ if $z_2^t \geq 1$. Since $z_2^t$ decreases by $\delta_1$ in each iteration, $x_{21}^t \neq x_{21}^{t+1}$ iff $z_2^{t+1} < 1$. However, since $z_2^t - z_2^{t+1} = \delta_1$, there can only be at most $\kappa_1 = \lceil 1/\delta_1 \rceil$ such $t$. For regions $Z_0$, $Z_2$, and $Z_3$, there exist similar $\kappa_0$, $\kappa_2$, and $\kappa_3$. Taking $\kappa = \max\{\kappa_0, \kappa_1, \kappa_2, \kappa_3\}$ completes the proof of the lemma. □

### F.3 Energy Increases by $\Theta(1)$ in Each Partition

Next, we show the energy in the system increases by a constant each time $z^t$ moves into a new partition. Again, we use this result in two places. First, we use the result in the proof of Lemma 9, to show that $z^t$ moves from $Z_i$ directly to $Z_{(i+2 \mod 4)}$ at most a constant number of times. Second, we use the result in combination with Lemma 9 to show $t_j \in \Theta(j^2)$ allowing us to conclude that $k \in \Theta(\sqrt{T})$ partitions are visited in $T$ iterations.

**Lemma 8.** $r_{j+1} - r_j \in \Theta(1)$.

The proof of Lemma 8 relies on the observation that (Gradient Descent) is simply a 1st order approximation of (Continuous Gradient Descent) as depicted in Figure 5. When neither $z_i^t \notin (0, 1)$, the continuous time dynamics move in a straight line and therefore a 1st order approximation perfectly preserves the energy of the system. However, if $z_i^t \in (0, 1)$ then by the strong convexity of $\bar{h}_i(z_i^t)$, the total energy of the system increases. By Lemma 7, there are a constant number of $t$ where $z_i^t \in (0, 1)$ for each partition and therefore the total energy increases by $O(1)$ in each partition.

Figure 5: Discrete Time is a 1st Order Approximation of Continuous Time.

For the proof of Lemma 8 is is useful to recall the following from Section 3.2:

$$\bar{h}_1^*(z_1^t) = \begin{cases} \alpha_{10} z_1^t - \beta_{10} & \text{if } z_1^t \leq 0 \\ \alpha_{11} z_1^t - \beta_{11} & \text{if } z_1^t \geq 1 \\ \gamma_1 (z_1^t)^2 + \alpha_1 z_1^t - \beta_1 & \text{otherwise} \end{cases} \tag{54}$$

$$\bar{h}_2^*(z_2^t) = \begin{cases} \alpha_{20} z_2^t - \beta_{20} & \text{if } z_2^t \leq 0 \\ \alpha_{21} z_2^t - \beta_{21} & \text{if } z_2^t \geq 1 \\ \gamma_2 (z_2^t)^2 + \alpha_2 z_2^t - \beta_2 & \text{otherwise} \end{cases} \tag{55}$$

where $\alpha_{i0} < 0$, $\alpha_{i1} > 0$, and $\gamma_i > 0$.

*Proof of Lemma 8.* Without loss of generality assume $z^{t_j}, ..., z^{t_{j+1}-1} \in Z_1$. Once again by selection of $Z_1$, $z_2^t > 0$ and $x_{11}^t = 1$ implying $z_1^t \geq 1$ for all $t = t_j, ..., t_{j+1} - 1$. Let $R^t = \sum_{i=1}^{2} \bar{h}_i^*(z_i^t)$ be the total energy in the system in iteration $t$. By Mertikopoulos et al. [2018], the continuous time dynamics are captured by $\{z : \sum_{i=1}^{2} h_i^*(z_i) = R\}$ around the point $z^t$. When $z_1^t \geq 1$, the continuous time dynamics around $z^t$ are captured by

$$R^t = \sum_{i=1}^{2} \bar{h}_i^*(z_i) \tag{56}$$

$$= \bar{h}_2^*(z_2) + \alpha_{11} z_1 - \beta_{11} \tag{57}$$

reducing to

$$z_1 = \frac{R^t + \beta_{11} - \bar{h}_2^*(z_2)}{\alpha_{11}}. \tag{58}$$

As observed earlier, (Gradient Descent) is simply a 1st order approximation of (Continuous Gradient Descent) and therefore

$$z_1^{t+1} = z_1^t - \frac{\nabla \bar{h}_2^*(z_2^t)}{\alpha_{11}}(z_2^{t+1} - z_2^t) \tag{59}$$

$$= z_1^t + \frac{\nabla \bar{h}_2^*(z_2^t)}{\alpha_{11}}\delta_1 \tag{60}$$

where $\delta_1 = z_2^t - z_2^{t+1}$ is shown to be constant in the proof of Lemma 7. We now examine the five possible locations for $z_2^t$ and $z_2^{t+1}$.

**Case 1:** $z_2^t \geq 1, z_2^{t+1} \geq 1$. We show there is no change to the energy in the system. Since $z_2^t \geq 1$,

$$z_1^{t+1} = z_1^t + \frac{\nabla \bar{h}_2^*(z_2^t)}{\alpha_{11}}\delta_1 \tag{61}$$

$$= z_1^t + \frac{\alpha_{21}}{\alpha_{11}}\delta_1 \tag{62}$$

The total energy in iteration $t+1$ is given by

$$R^{t+1} = \sum_{i=1}^{2} \bar{h}_i^*(z_i^{t+1}) \tag{63}$$

$$= \alpha_{11}z_1^{t+1} - \beta_{11} + \alpha_{21}z_2^{t+1} - \beta_{21} \tag{64}$$

$$= \alpha_{11}\left(z_1^t + \frac{\alpha_{21}}{\alpha_{11}}\delta_1\right) - \beta_{11} + \alpha_{21}(z_2^t - \delta_1) - \beta_{21} \tag{65}$$

$$= \alpha_{11}z_1^t - \beta_{11} + \alpha_{21}z_2^t - \beta_{21} \tag{66}$$

$$= \sum_{i=1}^{2} \bar{h}_i^*(z_i^t) = R^t \tag{67}$$

and the energy in the system remains unchanged.

**Case 2:** $z_2^t \in (0, 1), z_2^{t+1} \in (0, 1)$. We show the energy increases by at least $\gamma_2 \delta_1^2$. We begin with writing $z(\delta)$ as

$$z_1(\delta) = z_1^t + \frac{\nabla \bar{h}_2^*(z_2^t)}{\alpha_{11}}\delta \tag{68}$$

$$= z_1^t + \frac{2\gamma_2 z_2^t + \alpha_2}{\alpha_{11}}\delta_1 \tag{69}$$

$$z_2(\delta) = z_2^t - \delta \tag{70}$$

Therefore, $z^{t+1} = z(\delta_1)$. Similarly, let $R(\delta)$ be energy associated with the point $z(\delta)$. Formally,

$$R(\delta) = \sum_{i=1}^{2} \bar{h}_i^*(z_i^{t+1}(\delta)) \tag{71}$$

$$= \alpha_{11}z_1^{t+1}(\delta) - \beta_{11} + \gamma_2(z_2^{t+1}(\delta))^2 + \alpha_2 z_2(\delta) - \beta_2 \tag{72}$$

$$= \alpha_{11}\left(z_1^t + \frac{2\gamma_2 z_2^t + \alpha_2}{\alpha_{11}}\delta\right) - \beta_{11} + \gamma_2(z_2^t - \delta)^2 + \alpha_2(z_2^t - \delta) - \beta_2 \tag{73}$$

and $R(\delta_1) = R^{t+1}$ and $R(0) = R^t$. Moreover $\frac{d^2 R}{d\delta^2} = 2\gamma_2 > 0$ and therefore $R(\delta)$ is strongly convex with parameter $2\gamma_2$. Thus,

$$R^{t+1} = R(\delta_1) \geq R(0) + R'(0) + \gamma_2 \delta_1^2 \tag{74}$$

$$= R^t + \gamma_2 \delta_1^2 \tag{75}$$

and the energy increases by at least $\gamma_2 \delta_1^2$ completing Case 2.

**Case 3:** $z_2^t \geq 1, z_2^{t+1} \in (0, 1)$. The energy increases by at least $\gamma_2 (1 - z_2^{t+1})^2$. This case follows identically to Case 2 by approximating $R(\delta_1)$ using strong convexity and $R(z_2^t - 1)$.

**Case 4:** $z_2^t \in (0, 1), z_2^{t+1} \leq 0$. The energy increases by at least $\gamma_2 (z_2^t)^2$. This case follows similarly to Cases 2 and 3.

**Case 5:** $z_2^t \geq 1, z_2^{t+1} \leq 0$. The energy increases by at least $\gamma_2$. This case follows similarly to Cases 2-4.

We now can compute $r_{j+1} - r_j$. In each case, the increase in energy is bounded above since $z_2^t - z_2^{t+1}$ is bounded. Let $C_k$ be the number of times that Case $k$ occurs. Case 1 results in no change to the energy. By Lemma 7, Case 2 occurs at most $\kappa_1$ times. Since $z_2^t$ is decreasing, Cases 3, 4, and 5 can occur at most once each. Therefore $r_{j+1} - r_j \in \sum_{k=2}^{5} C_k \cdot O(1) \leq (\kappa_1 + 3) \cdot O(1) \in O(1)$. It remains to show $r_{j+1} - r_j \in \Omega(1)$.

First suppose Case 2 occurs at least once, then immediately we have $r_{j+1} - r_j \geq \gamma \delta_1^2 \in \Omega(1)$. If Case 2 does not occur, then either Cases 3 and 4 occur, or Case 5 occurs. If Case 5 occurs then $r_{j+1} - r_j \geq \gamma_2 \in \Omega(1)$ If Cases 3 and 4 occur but Case 2 does not, only one $t$ is such that $z_2^t \in (0, 1)$. Thus, $r_{j+1} - r_j \geq \min_{z_2 \in (0,1)} \{\gamma_2 (z_2)^2 + \gamma_2 (1 - z_2)^2\} = \frac{\gamma_2}{2} \in \Omega(1)$. In all possibilities, $r_{j+1} - r_j \in \Omega(1)$ completing the proof of the lemma. $\qquad\square$

## F.4 The Steps Per Partition are Proportional to the Energy

In this section, we show that the number of steps in a partition is proportional to the total energy in the system. We establish this by leveraging the connection between (Continuous Gradient Descent) and (Gradient Descent). Lemma 9 is used in conjunction with Lemma 8 to show a quadratic relationship between the total number of iterations and the number of partitions that the strategies have passed through. This quadratic relationship directly leads to the $O(\sqrt{T})$ regret bound in Theorem 3.

**Lemma 9.** $t_{j+1} - t_j \in \Theta(r_j)$.

*Proof of Lemma 9.* Without loss of generality, assume $z^{t_j}, ..., z^{t_{j+1}-1} \in Z_1$. As in the proof of Lemma 7, there exists a constant $\delta_1 > 0$ such that $z_2^t - z_2^{t+1} = \delta_1$ for all $t = t_j, ..., t_{j+1} - 1$. This implies $\delta_1 (t_{j+1} - t_j) = z_2^{t_j} - z_2^{t_{j+1}}$. Thus, to prove Lemma 9 it suffices to show $z_2^{t_j} - z_2^{t_{j+1}} \in \Theta(r_j)$. By definition of $t_{j+1}$, $z_2^{t_{j+1}-1} \in Z_1$ and therefore $0 \geq z_2^{t_{j+1}} = z_2^{t_{j+1}-1} - \delta \geq -\delta$. Thus, $z_2^{t_j} - z_2^{t_{j+1}} \in \Theta(r_j)$ if and only if $z_2^{t_j} \in \Theta(r_j)$.

To show $z_2^{t_j} \in \Theta(r_j)$ and complete the proof, we break the problem into 6 cases based on the location of $z^{t_j-1}$ as depicted in Figure 6. The analyses for Cases 1-3 are similar and we show Cases 4-6 can never occur.

**Case 1:** Let $R^{t_j-1}$ be the energy at time $t_j - 1$. Let $M_0 = \{z : z \in Z_0, z_1 \in [0, 1], \sum_{i=1}^{2} \bar{h}_i^*(z_i) = R^{t_j-1}\}$. By definition, $z^{t_j-1} \in M_0$. Observe for $z \in M_0$,

$$R^{t_j-1} = \sum_{i=1}^{2} \bar{h}_i^*(z_i) \tag{76}$$

$$= \gamma_1 (z_1)^2 + \alpha_1 z_1 - \beta_1 + \alpha_{21} z_2 - \beta_{21} \tag{77}$$

and therefore

$$z_2 = \frac{R^{t_j-1} - \gamma_1 (z_1)^2 - \alpha_1 z_1 + \beta_1 + \beta_{21}}{\alpha_{21}} \tag{78}$$

which is a concave function and therefore minimized at $z_1 = 0$ or $z_1 = 1$. Thus,

$$z_2^{t_j-1} \geq \min_{z_1 \in \{0,1\}} \frac{R^{t_j-1} - \gamma_1 (z_1)^2 - \alpha_1 z_1 + \beta_1 + \beta_{21}}{\alpha_{21}} \in \Theta(R^{t_j-1}). \tag{79}$$

Figure 6: Cases for Lemma 9.

Similar to the proof of Lemma 8, we compute $z_2^{t_j}$ from $z_2^{t_j-1}$:

$$z_2^{t_j} = z_2^{t_j-1} - \frac{\nabla \bar{h}_1^*(z_1^{t_j-1})}{\alpha_{21}}(z_1^{t_j} - z_1^{t_j-1}) \tag{80}$$

$$= z_2^{t_j-1} - \frac{2\gamma_1 z_1^{t_j-1} - \alpha_1}{\alpha_{21}}\delta_0 \tag{81}$$

$$\in z_2^{t_j-1} + \Theta(1) \in \Theta(R^{t_j-1}) \tag{82}$$

since $z_1^{t_j-1} \in [0,1]$. Finally, by Lemma 8, $r^{j-1} \leq R^{t_j-1} \leq r^j = r^{j-1} + \Theta(1)$ and therefore $z_2^{t_j} \in \Theta(r^j)$ completing Case 1.

**Case 2:** This case follows identically to Case 1 using $\nabla \bar{h}_1^*(z_1^{t_j-1}) = \alpha_{10}$.

**Case 3:** Similar to the proof of Case 1, $z_1^{t_j-1} \in -\Theta(R^{t_j-1})$ and $z_1^{t_j} = z_1^{t_j-1} + \Theta(1)$. However, since $z^{t_j} \in Z_1$, $z_1^{t_j} \geq 1$ implying $R^{t_j-1} \in \Theta(1)$. Therefore, by Lemma 8, $r^j \in \Theta(1)$. Let $\delta_3 > 0$ be as in the proof of Lemma 8. Since $z^{t_j-1} \in Z_3$ and $z_2^{t_j-1} \in [0,1]$, $z_2^{t_j} = z_2^{t_j-1} + \delta_3$ and $z_2^{t_j} \in \Theta(1) = \Theta(r^j)$ completing Case 3.

**Case 4, 5, and 6:** In Case 4, the sign of $\nabla \bar{h}_2^*(z_2^{t_j-1})$ implies $z_1^{t_j} < z_1^{t_j-1} < 0$. In Case 5, $z_1^{t_j} = z_1^{t_j-1} - \delta_2 < 1$ where $\delta_2 > 0$ is defined in the proof of Lemma 8. In Case 6, the sign of $\nabla \bar{h}_1^*(z_1^{t_j-1})$ implies $z_2^{t_j} < z_2^{t_j-1} < 0$. All three cases contradict that $z_1^{t_j} \in Z_1$ completing Cases 4, 5, and 6.

In all 6 cases, $z_2^{t_j} \in \Theta(r_j)$ implying $t_{j+1} - t_j \in \Theta(r_j)$ completing the proof. $\square$

# G   Convergence to the Boundary

*Proof of Theorem 1.*   The proof of convergence to the boundary follows similarly to the details for Theorem 3. By Bailey and Piliouras [2018], there exists a constant $w > 0$ and a $T$ such that $\min_{i \in \{1,2\}}\{|x_{i1}^t - x_{i1}^{NE}|\} \geq w$ for all $t \geq T$. Similar to Theorem 3, we can then partition the dual

space around the Nash equilibrium as follows:

$$Z_0 = \left\{ z : z_1 < x_{11}^{NE} + w, z_2 \geq x_{22}^{NE} + w \right\}.$$
$$Z_1 = \left\{ z : z_1 \geq x_{11}^{NE} + w, z_2 > x_{22}^{NE} - w \right\}.$$
$$Z_2 = \left\{ z : z_1 > x_{11}^{NE} - w, z_2 \leq x_{22}^{NE} - w \right\}.$$
$$Z_3 = \left\{ z : z_1 \leq x_{11}^{NE} - w, z_2 < x_{22}^{NE} + w \right\}.$$

Figure 7: Partitioning for Theorem 1.

Once again the strategies rotate clockwise when updated with (Gradient Descent). Similar to Lemma 8, the energy increases by at least a constant in each iteration. By continuity of $\bar{h}_i^*$ and compactness, the energy $\sum_{i=1}^{2} \bar{h}_i^*(z_i)$ is bounded above by $u$ when $z \in [0,1]^2$. Similar to Lemma 9, $z^t$ spends a bounded number of steps in a partition before moving onto the next partition. Since energy is increasing by a constant each time $z^t$ enters a new partition, there must exist an iteration $B$ when the energy exceeds $u$. Thus, for all $t \geq B$, $z^t \notin [0,1]^2$ implying $x^t$ is on the boundary. $\qquad\square$

## H    Proof of Theorem 5

In this section, we establish that the worst-case regret is exactly $\Theta(\sqrt{T})$. To establish this result, it remains to provide a game, learning rate, and initial condition $y^0$ where the regret is $\Omega(\sqrt{T})$. To establish this lower bound, we first express iteration $t$ uniquely with $t = \frac{n(n+1)}{2} + k$ for some $k \in \{0, ..., n\}$. Using notation, we provide the exact position of the payoff vector, $y_i^t$ in each iteration. With this position, we compute the exact utility and regret through iteration $t$. Specifically, we show that in iteration $\frac{n(n+1)}{2} + k$, the total regret is $\frac{n}{2} + O(1)$. To show these results, we use the game Matching Pennies with learning rate $\eta = 1$ and initial payoff vectors $y_1^0 = y_2^0 = (1, 0)$.

$$\begin{pmatrix} 1 & -1 \\ -1 & 1 \end{pmatrix} \qquad\qquad \text{(Matching Pennies Payoff Matrix)}$$

**Lemma 10.** *Consider the game Matching Pennies with learning rate $\eta = 1$ and initial conditions $y_1^0 = y_2^0 = (1, 0)$. In iteration $t = \frac{n(n+1)}{2} + k$ where $k \in \{0, ..., n\}$, player $i$'s payoff vector is given*

*by*

$$y_1^{\frac{n(n+1)}{2}+k} = \begin{cases} (1+k,-k) & \text{if } n \equiv 0 \mod 4 \\ (1+n-k,-n+k) & \text{if } n \equiv 1 \mod 4 \\ (-k,1+k) & \text{if } n \equiv 2 \mod 4 \\ (-n+k,1+n-k) & \text{if } n \equiv 3 \mod 4 \end{cases}$$

$$y_2^{\frac{n(n+1)}{2}+k} = \begin{cases} (1+n-k,-n+k) & \text{if } n \equiv 0 \mod 4 \\ (-k,1+k) & \text{if } n \equiv 1 \mod 4 \\ (-n+k,1+n-k) & \text{if } n \equiv 2 \mod 4 \\ (1+k,-k) & \text{if } n \equiv 3 \mod 4 \end{cases}.$$

*Proof.* The result trivially holds for the base case $t = n = k = 0$. We now proceed by induction and assume the results holds for $t = \frac{n(n+1)}{2} + k$ and show the result holds for $t+1$. We break the problem into four cases based on the remainder of $n/4$.

**Case 1:** $n \equiv 0 \mod 4$. By the inductive hypothesis, $y_1^t = (1+k,-k)$ and $y_2^t = (1+n-k,-n+k)$. Since $k \le n$, $y_{11}^t \ge 1$ and $y_{21}^t \ge 1$. Following similarly to Section 3.2,

$$x_{i1}^t = \begin{cases} 1 & \text{if } y_{i1}^t \ge 1 \\ 0 & \text{if } y_{i1}^t \le 0 \\ y_{i1}^t & \text{otherwise} \end{cases}. \tag{83}$$

Thus, $x_1^t = x_2^t = (1,0)$ implying

$$y_1^{t+1} = y_1^t + A x_2^t \tag{84}$$
$$= y_1^t + (1,-1) \tag{85}$$
$$= (1+k,-k) + (1,-1), \tag{86}$$
$$y_2^{t+1} = y_2^t - A^\mathsf{T} x_1^t \tag{87}$$
$$= y_2^t + (-1,1) \tag{88}$$
$$= (1+n-k,-n+k) + (-1,1). \tag{89}$$

If $k < n$, then $t+1 = \frac{n(n+1)}{2} + [k+1]$ and $y_1^{t+1} = (1+[k+1],-[k+1])$ and $y_2^{t+1} = (1+n-[k+1],-n+[k+1])$ as predicted by the statement of the lemma. If instead $k = n$, then $t+1 = \frac{[n+1]([n+1]+1)}{2}$ where $[n+1] \equiv 1 \mod 4$. Moreover, $y_1^{t+1} = (1+k+1,-k-1) = ([n+1]+1,-[n+1])$ and $y_2^{t+1} = (1+n-k-1,-n+k+1) = (0,1)$ again matching the statement of lemma. Thus, the inductive step holds for all values of $k$ when $n \equiv 0 \mod 4$.

**Case 2:** $n \equiv 1 \mod 4$. Since $k \in [0,n]$, $y_{11}^t \ge 1$ and $y_{21}^t \le 0$. Following identically to Case 1, $y_1^{t+1} = (1+n-k-1,-n+k+1)$ and $y_2^{t+1} = (-k-1,1+k+1)$ matching the statement of the lemma for all possible values of $k$.

**Case 3:** $n \equiv 2 \mod 4$. Since $k \in [0,n]$, $y_{11}^t \le 0$ and $y_{21}^t \le 0$. Following identically to the previous cases, $y_1^{t+1} = (-k-1,1+k+1)$ and $y_2^{t+1} = (-n+k+1,1+n-k-1)$ matching the statement of the lemma for all possible values of $k$.

**Case 4:** $n \equiv 3 \mod 4$. Since $k \in [0,n]$, $y_{11}^t \le 0$ and $y_{21}^t \ge 0$. Following identically to the previous cases, $y_1^{t+1} = (-n+k+1,1+n-k-1)$ and $y_2^{t+1} = (1+k+1,-k-1)$ matching the statement of the lemma for all possible values of $k$.

In all four cases, the inductive hypothesis holds completing the proof of the lemma. $\square$

With the exact value of the payoff vector in each iteration, we can compute the cumulative utility.

**Lemma 11.** *Consider the game Matching Pennies with learning rate $\eta = 1$ and initial conditions $y_1^0 = y_2^0 = (1,0)$. In iteration $t = \frac{n(n+1)}{2} + k$ where $k \in \{0,...,n\}$, player 1's cumulative utility is*

$$\sum_{s=0}^{t} x_1^s \cdot A x_2^s = \begin{cases} 1 - \frac{n}{2} + k & \text{if } n \equiv 0 \mod 2 \\ \frac{n-1}{2} - k & \text{if } n \equiv 1 \mod 2 \end{cases}.$$

*Proof.* We again proceed by induction. The base case $t = n = k = 0$ trivially holds. We assume the result holds for $t = \frac{n(n+1)}{2} + k$ and show it holds for $t+1$. Again, we break the problem into four cases based on the remainder of $n/4$.

**Case 1:** $n \equiv 0 \mod 4$. First, we consider $k < n$. Since $k < n$, $t+1$ is in the form $\frac{n(n+1)}{2} + [k+1]$ where $k+1 \leq n$. Thus, by Lemma 10, $y_{11}^{t+1} = 1 + [k+1] \geq 1$ and $y_{21}^{t+1} = 1 + n - [k+1] \geq 1$ implying $x_1^{t+1} = x_2^{t+1} = (1,0)$. therefore,

$$\sum_{s=0}^{t+1} x_1^s \cdot Ax_2^s = x_1^{t+1} \cdot Ax_2^{t+1} + \sum_{s=0}^{t} x_1^s \cdot Ax_2^s = 1 + 1 - \frac{n}{2} + k = 1 - \frac{n}{2} + [k+1]. \qquad (90)$$

This completes Case 1 when $k < n$.

If instead $k = n$, then $t$ is in the form $\frac{[n+1]([n+1]+1)}{2}$ where $[n+1] \equiv 1 \mod 4$. Similar to before, $y_{11}^{t+1} = 1 + n \geq 1$ and $y_{21}^{t+1} = 0$ implying $x_1^{t+1} = (1,0)$ and $x_2^{t+1} = (0,1)$. Therefore,

$$\sum_{s=0}^{t+1} x_1^s \cdot Ax_2^s = x_1^{t+1} \cdot Ax_2^{t+1} + \sum_{s=0}^{t} x_1^s \cdot Ax_2^s = -1 + 1 - \frac{n}{2} + n = \frac{[n+1]-1}{2}. \qquad (91)$$

This completes Case 1 when $k = n$. Thus the inductive hypothesis holds in Case 1.

**Case 2:** $n \equiv 2 \mod 4$. This case holds similarly to Case 1. The only difference is that $x_1^{t+1} = x_2^{t+1} = (0,1)$ when $k < n$ and $x_2^{t+1} = (1,0)$ when $k = n$ which does not change the value of $x_1^{t+1} \cdot Ax_2^{t+1}$.

**Case 3:** $n \equiv 1 \mod 4$. First consider $k < n$ implying $t+1$ is in the form $\frac{n(n-1)}{2} + [k+1]$ where $k+1 \leq n$. Similar to Case 1, $x_1^{t+1} = (1,0)$ and $x_2^{t+1} = (0,1)$. This implies

$$\sum_{s=0}^{t+1} x_1^s \cdot Ax_2^s = x_1^{t+1} \cdot Ax_2^{t+1} + \sum_{s=0}^{t} x_1^s \cdot Ax_2^s = -1 + \frac{n-1}{2} - k = \frac{n-1}{2} - [k+1]. \qquad (92)$$

completing Case 3 when $k < n$.

If instead $k = n$, then $t$ is in the form $\frac{[n+1]([n+1]+1)}{2}$ where $[n+1] \equiv 2 \mod 4$. This implies $x_1^{t+1} = (0,1)$ and $x_2^{t+1} = (0,1)$. Therefore,

$$\sum_{s=0}^{t+1} x_1^s \cdot Ax_2^s = x_1^{t+1} \cdot Ax_2^{t+1} + \sum_{s=0}^{t} x_1^s \cdot Ax_2^s = 1 + \frac{n-1}{2} - n = 1 - \frac{[n+1]}{2}, \qquad (93)$$

matching the statement of the lemma. This completes Case 3.

**Case 4:** $n \equiv 3 \mod 4$. Case 4 follows from Case 3 in the same way that Case 2 follows from Case 1. The hypothesis holds under all cases completing the proof of the lemma. $\qquad \square$

We now show that Matching Pennies with learning rate $\eta_1$ and initial conditions $y_1^0 = y_2^0 = (1,0)$ has regret $\Theta(\sqrt{T})$ when updated with (Gradient Descent).

*Proof of Theorem 5.* Theorem 3 establishes that the regret is $O(\sqrt{T})$. To show that the regret is $\Omega(\sqrt{T})$, we show that in iteration $t = \frac{n(n+1)}{2} + k$, that player 1's regret is $\frac{n}{2} + O(1)$ completing the proof.

The utility of the best fixed strategy through iteration $t$ is given by

$$\max_{x_1 \in \mathcal{X}_1} x_1 \cdot \sum_{s=0}^{t} Ax_2^s = \max_{x_1 \in \mathcal{X}_1} x_1 \cdot (y_1^{t+1} - y_1^0) \qquad (94)$$

$$\leq |y_{11}^{t+1}| + O(1) \qquad (95)$$

since $y_{11}^t - 1 = y_{12}^t$ for all $t$ by Lemma 10.

If $k < n$, then $t + 1 = \frac{n(n-1)}{2} + [k+1]$ and

$$|y_{11}^{t+1}| - \sum_{s=0}^{t} x_1^s \cdot A x_2^s = \begin{cases} [k+1] - (1 - \frac{n}{2} + k) = \frac{n}{2} & \text{if } n \equiv 0 \mod 4 \\ n - [k+1] - (\frac{n-1}{2} - k) = \frac{n-1}{2} & \text{if } n \equiv 1 \mod 4 \\ [k+1] + 1 - (1 - \frac{n}{2} + k) = \frac{n}{2} + 1 & \text{if } n \equiv 2 \mod 4 \\ n - [k+1] + 1 - (\frac{n-1}{2} - k) = \frac{n+1}{2} & \text{if } n \equiv 3 \mod 4 \end{cases}. \qquad (96)$$

If $k = n$, then $t + 1 = \frac{[n+1]([n+1]-1)}{2}$ and

$$|y_{11}^{t+1}| - \sum_{s=0}^{t} x_1^s \cdot A x_2^s = \begin{cases} [n+1] - (1 - \frac{n}{2} + n) = \frac{n}{2} & \text{if } n \equiv 0 \mod 4 \\ 1 - (\frac{n-1}{2} - n) = \frac{n+3}{2} & \text{if } n \equiv 1 \mod 4 \\ [n+1] + 1 - (1 - \frac{n}{2} + n) = \frac{n}{2} + 1 & \text{if } n \equiv 2 \mod 4 \\ 0 - (\frac{n-1}{2} - n) = \frac{n+1}{2} & \text{if } n \equiv 3 \mod 4 \end{cases}. \qquad (97)$$

In all cases, the total regret is $\frac{n}{2} + O(1) \in \Omega(\sqrt{t})$ completing the proof of the theorem. □

# I   Full Results of Experiments

An alternative proof of Theorem 3 can be described geometrically as follows: In iteration $t$ of (Gradient Descent), the payoff vectors are contained on some 2-dimensional ball in the dual space. Since (Gradient Descent) updates strategies myopically, the payoff vectors cycle along this 2-dimensional ball. In most iterations, the size of the ball increases. Since the change in player 1's (player 2's) payoff vector is bounded by the rows (respectively columns) of the payoff matrix $A$, the increasing size of the ball implies that the time to complete a cycle grows over time.

Players' strategies are obtained by projecting the payoff vectors into the simplex. As a result, player strategies also cycle. Moreover, the time to complete a cycle also grows over time implying that the average step-size goes to zero. Specifically, in the proof of Theorem 3, $\sum_{t=1}^{T} ||x_1^{t+1} - x_1^t|| \in O(\sqrt{T})$ implying sublinear regret.

In higher dimensions, we can make a similar argument to suggest sublinear regret. Once again, in any given iteration the payoff vectors are on the surface of some high dimensional ball in the dual space. By definition, the $j^{th}$ component of $h^*(y_i)$ is strictly convex whenever $x_{ij} \in (0,1)$. As a result, the ball containing the payoff vector increases in any iteration where there is a $j$ such that $\{x_{ij}^t, x_{ij}^{t+1}\} \in (0,1)^2$. In addition, the myopic nature of (Gradient Descent) suggests that the payoff vectors will not be contained to any small section of the ball in the dual space, e.g., the strategies cycle. As such, we would expect that the average step-size in the strategy space again goes to zero implying sublinear regret. Thus, we conjecture that the regret is sublinear in higher dimensions.

To test this conjecture, we generated 30 random payoff matrices for 2x2, 5x5, 10x10, and 50x50 games and updated strategies in these games with (Gradient Descent) for 10,000 iterations. The payoff matrix $A$ was generated uniformly at random from $[0,10]^{n \times n}$ for $n \in \{2, 5, 10, 50\}$. The initial payoff vector $y_i^0$ was generated uniformly at random from $[0,1]^n$. Agents' strategies were then updated via (Gradient Descent) for 10,000 iterations with learning rate $\eta = 1$. In our experiments, we exclude payoff matrices that have a single pure strategy Nash equilibria – Bailey and Piliouras [2018]'s analysis can be extended to show (Gradient Descent) will converge to the Nash equilibrium in these settings and the regret will be bounded. Experiments were conducted with the statistical software R and the source code is available at http://www.jamespbailey.com/FastAndFuriousGD.html.

We tested how well $b \cdot T^a$ predicts regret for (Gradient Descent), via a logarithmic regression with the model $\log Regret_1(T) \approx a \cdot \log T + \log b$. A summary of the result of the regression can be found in Table 2.

For all 120 of our instances, our logarithmic regression estimates that regret grows significantly slower than $O(T)$ with most instances obtaining regret close to $O(\sqrt{T})$. Moreover, these models explain nearly all of the variability in regret. Estimates for individual replicates are given in Tables 4-7.

In the set of 5x5 games, replicates 13 and 27 appear to have much lower regret that $O(\sqrt{T}) - T^{.3262}$ and $T^{.3323}$ respectively. To see if there are indeed instances where regret is significantly lower than

Table 2: Regression Summary for 10,000 Iterations of Gradient Descent in 30 Random Games

| strategies | $Regret_1(T) \approx b \cdot T^a$ | p-value | % of variability explained | $\lvert$support of $x^*\rvert$ |
|---|---|---|---|---|
| 2 | $a \in [0.4151, 0.5497]$ | $< .000001$ | $90.59 - 99.88$ | 2 |
| 5 | $a \in [0.3262, 0.5372]$ | $< .000001$ | $93.00 - 99.81$ | 2-5 |
| 10 | $a \in [0.4012, 0.5490]$ | $< .000001$ | $99.10 - 99.80$ | 2-9 |
| 50 | $a \in [0.5195, 0.5856]$ | $< .000001$ | $99.45 - 99.89$ | 18-29 |

$O(\sqrt{T})$, we further simulated these two replicates for a total of 500,000 iterations and summarized the results in Table 3. By increasing the number of iterates, we observe that both replicates exhibit regret closer to $\sqrt{T}$ as expected from our theoretical results.

Table 3: 500,000 Iterations of Gradient Descent for Replicates 13 and 27 in 5x5 Games

| replicate | $Regret_1(T) \approx b \cdot T^a$ | p-value | % of variability explained | $\lvert$support of $x^*\rvert$ |
|---|---|---|---|---|
| 13 | $a = .4481$ | $< .000001$ | 98.89 | 2 |
| 27 | $a = .4666$ | $< .000001$ | 99.23 | 2 |

Table 4: Regression for 30 Replicates of Gradient Descent in 2x2 Games

| replicate | $Regret_1(T) \approx b \cdot T^a$ | p-value | % of variability explained | $\lvert$support of $x^*\rvert$ |
|:---:|:---:|:---:|:---:|:---:|
| 1 | $a = 0.5423$ | $< .000001$ | 98.93 | 2 |
| 2 | $a = 0.4981$ | $< .000001$ | 99.60 | 2 |
| 3 | $a = 0.5083$ | $< .000001$ | 99.77 | 2 |
| 4 | $a = 0.5073$ | $< .000001$ | 99.55 | 2 |
| 5 | $a = 0.4847$ | $< .000001$ | 98.90 | 2 |
| 6 | $a = 0.4949$ | $< .000001$ | 99.54 | 2 |
| 7 | $a = 0.5291$ | $< .000001$ | 99.74 | 2 |
| 8 | $a = 0.4958$ | $< .000001$ | 99.18 | 2 |
| 9 | $a = 0.4739$ | $< .000001$ | 99.76 | 2 |
| 10 | $a = 0.4992$ | $< .000001$ | 99.88 | 2 |
| 11 | $a = 0.5143$ | $< .000001$ | 98.83 | 2 |
| 12 | $a = 0.4871$ | $< .000001$ | 99.16 | 2 |
| 13 | $a = 0.5269$ | $< .000001$ | 99.15 | 2 |
| 14 | $a = 0.5363$ | $< .000001$ | 99.74 | 2 |
| 15 | $a = 0.5086$ | $< .000001$ | 99.63 | 2 |
| 16 | $a = 0.5497$ | $< .000001$ | 99.17 | 2 |
| 17 | $a = 0.5361$ | $< .000001$ | 99.52 | 2 |
| 18 | $a = 0.5058$ | $< .000001$ | 99.82 | 2 |
| 19 | $a = 0.4911$ | $< .000001$ | 99.49 | 2 |
| 20 | $a = 0.5147$ | $< .000001$ | 99.70 | 2 |
| 21 | $a = 0.4152$ | $< .000001$ | 90.59 | 2 |
| 22 | $a = 0.5097$ | $< .000001$ | 99.73 | 2 |
| 23 | $a = 0.5276$ | $< .000001$ | 99.57 | 2 |
| 24 | $a = 0.4626$ | $< .000001$ | 99.36 | 2 |
| 25 | $a = 0.5004$ | $< .000001$ | 99.53 | 2 |
| 26 | $a = 0.5020$ | $< .000001$ | 99.81 | 2 |
| 27 | $a = 0.4933$ | $< .000001$ | 98.79 | 2 |
| 28 | $a = 0.4938$ | $< .000001$ | 99.63 | 2 |
| 29 | $a = 0.5078$ | $< .000001$ | 99.45 | 2 |
| 30 | $a = 0.5013$ | $< .000001$ | 98.99 | 2 |

Table 5: Regression for 30 Replicates of Gradient Descent in 5x5 Games

| replicate | $Regret_1(T) \approx b \cdot T^a$ | p-value | % of variability explained | \|support of $x^*$\| |
|---|---|---|---|---|
| 1 | $a = 0.4042$ | $< .000001$ | 97.98 | 2 |
| 2 | $a = 0.4965$ | $< .000001$ | 99.14 | 4 |
| 3 | $a = 0.4654$ | $< .000001$ | 98.95 | 3 |
| 4 | $a = 0.4899$ | $< .000001$ | 99.24 | 2 |
| 5 | $a = 0.4402$ | $< .000001$ | 99.13 | 2 |
| 6 | $a = 0.4954$ | $< .000001$ | 99.67 | 2 |
| 7 | $a = 0.4602$ | $< .000001$ | 99.64 | 3 |
| 8 | $a = 0.4941$ | $< .000001$ | 99.00 | 3 |
| 9 | $a = 0.4484$ | $< .000001$ | 99.56 | 3 |
| 10 | $a = 0.5119$ | $< .000001$ | 99.72 | 5 |
| 11 | $a = 0.4818$ | $< .000001$ | 99.65 | 2 |
| 12 | $a = 0.5092$ | $< .000001$ | 99.56 | 3 |
| 13 | $a = 0.3262$ | $< .000001$ | 93.59 | 2 |
| 14 | $a = 0.4814$ | $< .000001$ | 99.63 | 2 |
| 15 | $a = 0.4689$ | $< .000001$ | 99.81 | 2 |
| 16 | $a = 0.4520$ | $< .000001$ | 99.73 | 3 |
| 17 | $a = 0.5372$ | $< .000001$ | 99.50 | 3 |
| 18 | $a = 0.4134$ | $< .000001$ | 98.14 | 3 |
| 19 | $a = 0.4653$ | $< .000001$ | 99.21 | 3 |
| 20 | $a = 0.4962$ | $< .000001$ | 97.38 | 2 |
| 21 | $a = 0.5129$ | $< .000001$ | 99.62 | 2 |
| 22 | $a = 0.4932$ | $< .000001$ | 99.39 | 3 |
| 23 | $a = 0.4534$ | $< .000001$ | 99.37 | 2 |
| 24 | $a = 0.4839$ | $< .000001$ | 99.40 | 3 |
| 25 | $a = 0.5168$ | $< .000001$ | 99.74 | 3 |
| 26 | $a = 0.5216$ | $< .000001$ | 99.33 | 4 |
| 27 | $a = 0.3323$ | $< .000001$ | 93.00 | 2 |
| 28 | $a = 0.4058$ | $< .000001$ | 99.05 | 3 |
| 29 | $a = 0.5269$ | $< .000001$ | 99.28 | 3 |
| 30 | $a = 0.5234$ | $< .000001$ | 99.57 | 4 |

Table 6: Regression for 30 Replicates of Gradient Descent in 10x10 Games

| replicate | $Regret_1(T) \approx b \cdot T^a$ | p-value | % of variability explained | \|support of $x^*$\| |
|---|---|---|---|---|
| 1 | $a = 0.5490$ | $< .000001$ | 99.76 | 7 |
| 2 | $a = 0.4953$ | $< .000001$ | 99.47 | 7 |
| 3 | $a = 0.4682$ | $< .000001$ | 99.12 | 6 |
| 4 | $a = 0.4707$ | $< .000001$ | 99.63 | 5 |
| 5 | $a = 0.5017$ | $< .000001$ | 99.73 | 5 |
| 6 | $a = 0.5262$ | $< .000001$ | 99.80 | 7 |
| 7 | $a = 0.5323$ | $< .000001$ | 99.67 | 5 |
| 8 | $a = 0.4863$ | $< .000001$ | 99.64 | 3 |
| 9 | $a = 0.4903$ | $< .000001$ | 99.71 | 4 |
| 10 | $a = 0.5363$ | $< .000001$ | 99.45 | 7 |
| 11 | $a = 0.4875$ | $< .000001$ | 99.65 | 5 |
| 12 | $a = 0.4921$ | $< .000001$ | 99.51 | 5 |
| 13 | $a = 0.4958$ | $< .000001$ | 99.12 | 4 |
| 14 | $a = 0.5341$ | $< .000001$ | 99.53 | 7 |
| 15 | $a = 0.5209$ | $< .000001$ | 99.20 | 4 |
| 16 | $a = 0.5044$ | $< .000001$ | 99.66 | 5 |
| 17 | $a = 0.5185$ | $< .000001$ | 99.80 | 5 |
| 18 | $a = 0.4900$ | $< .000001$ | 99.71 | 6 |
| 19 | $a = 0.5063$ | $< .000001$ | 99.39 | 6 |
| 20 | $a = 0.5356$ | $< .000001$ | 99.78 | 6 |
| 21 | $a = 0.5232$ | $< .000001$ | 99.79 | 9 |
| 22 | $a = 0.4967$ | $< .000001$ | 99.45 | 5 |
| 23 | $a = 0.5113$ | $< .000001$ | 99.65 | 5 |
| 24 | $a = 0.5137$ | $< .000001$ | 99.10 | 5 |
| 25 | $a = 0.5103$ | $< .000001$ | 99.67 | 5 |
| 26 | $a = 0.5348$ | $< .000001$ | 99.29 | 7 |
| 27 | $a = 0.4973$ | $< .000001$ | 99.55 | 7 |
| 28 | $a = 0.4798$ | $< .000001$ | 99.58 | 4 |
| 29 | $a = 0.4012$ | $< .000001$ | 99.19 | 2 |
| 30 | $a = 0.4956$ | $< .000001$ | 99.38 | 6 |

Table 7: Regression for 30 Replicates of Gradient Descent in 50x50 Games

| replicate | $Regret_1(T) \approx b \cdot T^a$ | p-value | % of variability explained | \|support of $x^*$\| |
|---|---|---|---|---|
| 1 | $a = 0.5544$ | $< .000001$ | 99.76 | 20 |
| 2 | $a = 0.5340$ | $< .000001$ | 99.78 | 24 |
| 3 | $a = 0.5318$ | $< .000001$ | 99.82 | 23 |
| 4 | $a = 0.5309$ | $< .000001$ | 99.68 | 20 |
| 5 | $a = 0.5323$ | $< .000001$ | 99.61 | 18 |
| 6 | $a = 0.5433$ | $< .000001$ | 99.70 | 27 |
| 7 | $a = 0.5781$ | $< .000001$ | 99.84 | 21 |
| 8 | $a = 0.5442$ | $< .000001$ | 99.77 | 20 |
| 9 | $a = 0.5584$ | $< .000001$ | 99.65 | 26 |
| 10 | $a = 0.5425$ | $< .000001$ | 99.85 | 23 |
| 11 | $a = 0.5233$ | $< .000001$ | 99.54 | 20 |
| 12 | $a = 0.5675$ | $< .000001$ | 99.67 | 23 |
| 13 | $a = 0.5703$ | $< .000001$ | 99.45 | 23 |
| 14 | $a = 0.5471$ | $< .000001$ | 99.72 | 29 |
| 15 | $a = 0.5297$ | $< .000001$ | 99.51 | 22 |
| 16 | $a = 0.5233$ | $< .000001$ | 99.75 | 21 |
| 17 | $a = 0.5792$ | $< .000001$ | 99.61 | 23 |
| 18 | $a = 0.5457$ | $< .000001$ | 99.69 | 21 |
| 19 | $a = 0.5227$ | $< .000001$ | 99.72 | 20 |
| 20 | $a = 0.5203$ | $< .000001$ | 99.64 | 24 |
| 21 | $a = 0.5459$ | $< .000001$ | 99.72 | 23 |
| 22 | $a = 0.5368$ | $< .000001$ | 99.73 | 24 |
| 23 | $a = 0.5457$ | $< .000001$ | 99.59 | 25 |
| 24 | $a = 0.5856$ | $< .000001$ | 99.48 | 23 |
| 25 | $a = 0.5624$ | $< .000001$ | 99.83 | 25 |
| 26 | $a = 0.5394$ | $< .000001$ | 99.70 | 20 |
| 27 | $a = 0.5501$ | $< .000001$ | 99.89 | 27 |
| 28 | $a = 0.5444$ | $< .000001$ | 99.73 | 25 |
| 29 | $a = 0.5195$ | $< .000001$ | 99.71 | 23 |
| 30 | $a = 0.5552$ | $< .000001$ | 99.84 | 27 |