[Reviews · NeurIPS 2019]

Reviewer 1



This paper studies gradient descent with a fixed step size for 2x2 zero-sum games. The key theoretical result is that while in discrete time, FTRL only guarantees linear regret for a fixed step size, by utilizing the geometry in gradient descent, one may obtain sublinear regret in 2x2 zero-sum games. The authors show that after a long enough time, strategies will lie on the boundary of the strategy space. The main result (Theorem 2) is obtained by characterizing the exact dynamics for gradient descent, suitably selecting a dual space, decomposing each “rotation” into distinct components, analyzing the change in energy in each iteration for each component and showing that time is quadratic in the number of partitions (rotations). The authors give an example which satisfies a matching lower bound, showing that their analysis is tight. Lastly, the authors demonstrate through simulations that the regret bound may possibly hold for larger zero-sum games. Clarity: The paper is generally easy to follow at a high level, with most details being deferred to the appendix. I found Figures 2 and 3 to be particularly useful in providing intuition to the analysis. The authors should be commended for putting the effort in making their analysis accessible. However, Section 3 and 3.1 could be further improved. In particular, it seems that the result in Lemma 6 is a prerequisite to understanding expressions (5)-(9)--- in fact, the set S_i is not even defined in the main paper at that point. Section 3 would have been much easier to read had the result from Lemma 6 been included while omitting the derivations from (5)-(8). A few very minor comments: (1) The phrase “This is because y^{t+1}_i cannot appear anywhere in R^{n_i}” can be a little confusing at first. (2) In Figures 2 and 3, the term “Payoff vector” (previously described by y) is overloaded and used to describe the “Transformed payoff vector” z. (3) In Figure 3, the strategies x^t are not visible. The label could be removed if player strategies are not central to the point being illustrated by the figure. Quality: I did not go through every derivation in the appendix, especially when it came to each of the cases in Lemmas 8 and 9. The proof sketches in the main paper, as well as the key ideas in Appendix C, D, E, and F seem correct, at least to the best of my ability. Significance: This analysis in this paper should be of interest to the online learning and optimization communities. Given that 2x2 zero-sum games are just about the simplest games possible, I am doubtful that this paper will be of direct interest to researchers outside these niche communities. Nonetheless, it is possible that the analysis could be a useful springboard for future research. Originality The paper builds upon the techniques utilized by Mertikopoulos et al [2018] and Bailey and Piliouras [2018, 2019]. The geometric analysis (restricted to 2x2 games) and key results proven in the paper are novel as far as I can tell. ============================================= Post-rebuttal: I am satisfied with the authors' response. That said, the authors' should include their rebuttal regarding extensions to higher dimensions and FTRL, together with the inclusion of Lemma 6 if the paper is accepted.

Reviewer 2



This paper studies gradient descent learning with fixed step sizes in two player, two strategy zero sum games. This paper makes several contributions and the results are obtained via a careful analysis of the dynamics that is rather insightful. Moreover, the paper is clearly written and polished. Several interesting examples are given that are valuable to the exposition of the paper. The main result of the paper is showing that online gradient descent achieves sub-linear regret with fixed step sizes in two player two strategy zero sum games. A matching lower bound on the regret is given, showing the main result is tight. This result is significant, given that such a result has not been obtained for fixed step sizes and without knowledge of the time horizon. This work built on a lot of recent works to obtain the results and the techniques are novel in my opinion. I believe that the deep understanding of the problem demonstrated in this work will be useful for future works. The primary drawbacks I could see about the paper is the limited scope of the problem and in some sense the objective. In particular, since the paper only focuses on two player, two strategy games, the applications are limited and it is primary theoretical contribution. However, this is where the field is now and obtaining a strong understanding of simple games will help the study of general sum games and more complicated action spaces. There is some question in my opinion on the significance of time-averaged play converging to a Nash equilibrium when the empirical play does not. I would only suggest that the authors provide some discussion in the paper on why it is desirable for time-averaged play to converge to Nash when the empirical play does not. The simulations in higher dimensions are interesting and simulating outside of what can be proved gives some intuition about what stronger results could be obtained. However, I am not sure it should be claimed that the experimental evidence demonstrates stronger regret bounds hold for all zero-sum games. The games that are simulated are rather benign, and as a result, I am not convinced the results indicate that it is likely results extend to higher dimensions. For this reason, I suggest the authors weaken the claims in this respect. I have a few suggestions that may improve readability, if space permits. Lemma 6 is mentioned several times in the main body of the paper, without a statement of the result or much explaining it and without reference to where it can be found. I suggest the authors consider moving this result into the main paper if they can find space, and if they cannot, I suggest given a brief explanation of the claim and pointing to where it can be found in the appendix. Along these lines, the authors may consider giving a brief explanation on page 5 of the maximizing argument in Kakade, 2009. --------------------------- Following author response ---------------------------- The points the authors make in the response are interesting and would certainly help future work extend this paper. If the authors feel comfortable sharing this information regarding challenges proving results for higher dimensions and other algorithms, I would encourage them to include much of what was in the response in a final version. Since the authors did not address it in the response, I want to reiterate that the authors may consider stating or giving an explanation of lemma 6 in the main paper since I was not the only reviewer who mentioned that it may improve readability.

Reviewer 3



This paper considers the use of online learning algorithms in min-max optimization. Classic results show that no-regret algorithms such as Online Gradient Descent can achieve $O(\sqrt{T})$ regret when faced with adversarial loss functions. Moreover, in two-player zero-sum games where both players use a no-regret algorithm, the average iterates of the two players will converge to a Nash Equilibrium. However, these classic results require adaptive step-sizes or step-sizes of roughly $1/\sqrt{T}$. That is, there are no fixed-step size algorithms that simultaneously get regret bounds for all $T$. This paper addresses this gap in the literature by proving several results about the gradient descent/ascent dynamics for two-player two-strategy zero-sum games with any fixed step-size. In particular, the authors show that the regret bound of each player in this setting is $\Theta(\sqrt{T})$, which implies that the time-average of the players' strategies converges to the Nash Equillibrium at a rate of $\Theta(\sqrt{T})$. The analysis focuses largely on characterizing the evolution of the payoff vectors for both players over time. Since the analysis only focuses on the 2x2 payoff matrix case, for each player the authors can write down the payoff of the first action in terms of the second action, which allows for a simplified representation of the evolution of the game. They show that the dynamics move to the boundary of the space and cycle between different regions along the boundary. The analysis follows by tracking a certian potential function and carefully tracking how this potential function changes in each region in addition to tracking how long the dynamics stay in each region. The strength of this paper is that it addresses an important gap in the understanding of a classic online learning algorithm. While online learning algorithms were originally designed to work in the presence of adversarial loss functions, much recent work has focused on gaining a better understanding of these algorithms in settings that aren't compeltely adversarial, such as finding Nash Equilibria or nonconvex optimization. This work fits into that vein of work nicely. The weakness of this paper is that it only gives results for the two-dimensional case. Moreover, the analysis seems fairly specialized to the two-dimensional case since it relies on a precise characterization of the trajectory of the algorithm, which seems difficult to write down explicitly for the general n-dimensional case. Overall, this paper seems like a weak accept to me. I think the work identifies an important question, namely "Can gradient descent/ascent with constant step-size find Nash Equilibria?" Much of the analysis consists of straightforward calculations, but the overall approach of characterizing the different phases of the dynamics seems fairly original for this setting and is a nontrivial contribution. The paper is also fairly clear. These factors make the paper seem like a weak accept even though the contribution is in a very limited setting. ====== Post-rebuttal ====== I read the rebuttal and other reviews. I am fine with accepting this paper if the authors add 1) an appendix on extensions to other FTRL algorithms and 2) the discussion from the rebuttal about obstacles to proving the result in higher-dimensions.

[Author Response · NeurIPS 2019]

We thank the reviewers for their excellent feedback. All the comments showed a strong understanding of the paper and will be useful for presenting our results in the best form. We agree with all suggested minor changes and will update the paper accordingly. The remaining comments focus mainly on an interest in a higher dimensional argument and other algorithms.

**Higher Dimensions:** Reviewer 2 and 3 asked about extending the proof to higher dimensions. Our proof consists of the three components listed below.

(1) the "step-size" in the dual space is bounded; i.e., $a \leq ||y_i^t - y_i^{t-1}|| \leq b$.

(2) a proof of divergence in the dual/payoff space where the divergence grows linearly when at least one agent is not playing a pure strategy and negligibly when both agents are playing a pure strategy.

(3) a proof of recurrence where the "cycle" length (in the strategy/primal space) is bounded.

The first two components immediately extend using our current analysis. In regards to the last step, recent advancements in understanding the geometry of learning dynamics in larger games (e.g., Mertikopoulos et al, 2018; Bailey and Piliouras, 2019) suggests that although non-trivial this last step can also be eventually rigorously established. New ideas, however, will most likely be needed for this last step. Given our framework, we believe that as more techniques are developed to understand these trajectories, sublinear regret will readily follow using our approach.

**Other Algorithms:** Reviewer 3 expresses an interest in analysis of algorithms similar to gradient descent. Reviewer 3's hunch that a similar analysis could work on other algorithms is on point. Our proof technique should extend to all FTRL algorithms via the dual space discussed in our paper. Both (1) and (3) trivially extend to this framework. As discussed in Appendix I, the proof for (2) mainly uses the strict convexity of the regularizer in FTRL. We opted to focus this paper on gradient descent because the core principles behind the proofs are most clear when working with the regularizer $h(x) = ||x||^2$. In particular, GD has two nice properties:

(a) the dual space is the affine hull of the strategy space and the mapping from the dual space to the primal space is a simple projection. This property isn't particularly useful in the proofs. Rather, the dual space is still an underutilized concept and we believe the techniques are most accessible and most likely to be reused if there is a simple connection between the two spaces.

(b) after a finite number of iterations, all primal strategies appear on the boundary. This does not hold for all FTRL algorithms (e.g., MWU always selects fully-mixed strategies). For other FTRL algorithms and for any $\epsilon$, after a finite number of iterations all strategies will appear within $\epsilon$ of the boundary. The proof of this follows identically to the proof in Appendix G. However, to establish linear growth in divergence as in Appendix E.3., $\epsilon$ will likely have to be carefully selected for each cycle. This is because for an algorithm like MWU, the convex conjugate $h^*$ is never linear; rather it becomes arbitrarily close to a linear function as both agents come closer to playing a pure strategy.

We are certainly open to including an additional appendix to discuss the extension to FTRL. However, we strongly believe that the paper is best presented through the lens of gradient descent. By taking advantage of (a) and (b) while introducing our techniques, we believe researchers will find our analysis more accessible and will be more likely use similar ideas to advance the understanding of online optimization.

**Other comments:** On a last note, we agree with reviewer 2 that experiments are only suggestive that the stronger regret bounds extend to larger zero-sum games. We will make sure to make that point more clear.

[Meta-Review · NeurIPS 2019]

While there was consensus on acceptance, reviewers made several important requests to add detail, including material from the author response; please see the updated reviews. In brief, please add a discussion of challenges in higher dimensions, an explanation of Lemma 6, and a discussion of other FTRL algorithms.